# Deep Learning Approaches for Video Compression: A Bibliometric Analysis

**Ranjeet Vasant Bidwe** [1,*] **, Sashikala Mishra** [1,*] **, Shruti Patil** [2] **, Kailash Shaw** [1] **, Deepali Rahul Vora** [1] **, Ketan Kotecha** [2] **and Bhushan Zope** [1]

1 Symbiosis Institute of Technology, Symbiosis International (Deemed University) (SIU), Lavale, Pune 412115, India; kailash.shaw@gmail.com (K.S.); deepali.vora@sitpune.edu.in (D.R.V.); bhushan.zope.phd2021@sitpune.edu.in (B.Z.)

2 Symbiosis Centre for Applied Artificial Intelligence (SCAAI), Symbiosis Institute of Technology, Symbiosis International (Deemed University) (SIU), Lavale, Pune 412115, India; shruti.patil@sitpune.edu.in (S.P.); director@sitpune.edu.in (K.K.)

* Correspondence: ranjeet.bidwe.phd2021@sitpune.edu.in (R.V.B.); sashikala.mishra@sitpune.edu.in (S.M.)

**Abstract:** Every data and kind of data need a physical drive to store it. There has been an explosion in the volume of images, video, and other similar data types circulated over the internet. Users using the internet expect intelligible data, even under the pressure of multiple resource constraints such as bandwidth bottleneck and noisy channels. Therefore, data compression is becoming a fundamental problem in wider engineering communities. There has been some related work on data compression using neural networks. Various machine learning approaches are currently applied in data compression techniques and tested to obtain better lossy and lossless compression results. A very efficient and variety of research is already available for image compression. However, this is not the case for video compression. Because of the explosion of big data and the excess use of cameras in various places globally, around 82% of the data generated involve videos. Proposed approaches have used Deep Neural Networks (DNNs), Recurrent Neural Networks (RNNs), and Generative Adversarial Networks (GANs), and various variants of Autoencoders (AEs) are used in their approaches. All newly proposed methods aim to increase performance (reducing bitrate up to 50% at the same data quality and complexity). This paper presents a bibliometric analysis and literature survey of all Deep Learning (DL) methods used in video compression in recent years. Scopus and Web of Science are well-known research databases. The results retrieved from them are used for this analytical study. Two types of analysis are performed on the extracted documents. They include quantitative and qualitative results. In quantitative analysis, records are analyzed based on their citations, keywords, source of publication, and country of publication. The qualitative analysis provides information on DL-based approaches for video compression, as well as the advantages, disadvantages, and challenges of using them.

**Keywords:** video compression; image compression; deep neural networks

## 1. Introduction

Most data generated in the world today is videos [1–3]. The main task given to compression techniques is to minimize the number of bits required for code on data or information provided, further minimizing the memory required to store the given data. Graceful degradation is a quality-of-service term used to explain that as bandwidth drops or transmission error occurs, the user experience becomes degraded and tries to be meaningful. Traditional data compression algorithms use handcrafted encoder–decoder pairs called "codecs". The main problem is adaptability, and they are not sure whether data are being compressed or degraded gracefully. These techniques have been developed for bitmap images (images organized as a grid of color points called Pixels). They cannot be extended to various new media formats such as 360 videos. Moreover, compression is

necessary for much real-time and complex applications such as space, live time-series data, and medical imaging, which require exact recoveries of original images. Many human efforts are spent analyzing the details of these new data formats and providing efficient compression methods. Therefore, there is a need for new data compression algorithms that will increase flexibility while demonstrating improvements on traditional measures of compression quality.

There has been a significant evolution in the field of data compression in the past few decades. Data compression can be categorized into Lossless and Lossy, as shown in Figure 1. Moreover, a new kind of compression type called Near lossless compression is currently being supported by newly proposed compression techniques. If you are using lossless compression, the picture quality will remain the same, and you can obtain an image or video of original size and quality after decompression. Lossy compression tries to find redundant pixel information and permanently removes it from the original image. Thus, lossy compression is not used for compressing text documents or software but is widely used for media elements such as audio, video, or images. Lossy compression algorithms take advantage of the inherent limitations of the human eye and discard information that can be seen. JPEG (Joint Photographic Expert Group) [4], JPEG2000 [5], BPG (Better Portable Graphics), and MPEG (Motion Picture Experts Group) [6], including MPEG CDVS (MPEG Compact Descriptors for Visual Search) [7], and MPEG CDVA (MPEG Compact Descriptors for Visual Analysis) [7] and MP3 [8] are current software that are using lossy compression. On the other hand, file data are temporarily thrown away to transfer files over the internet in lossless compression. It can be applied to graphics and computer data such as spreadsheets, text documents, or software. Portable Network Graphics (PNG) [9], windows tool WINZIP [10], and GNU tool gzip [11] all use lossless compression.

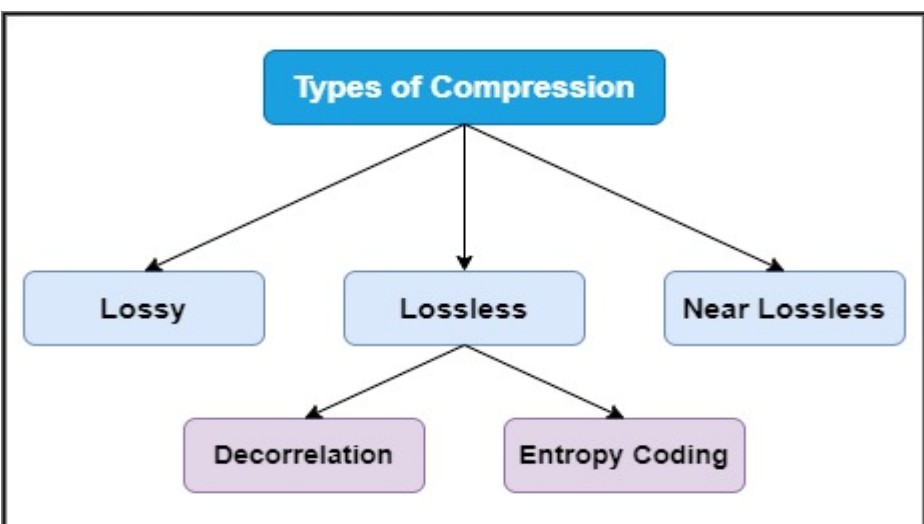

**Figure 1.** Types of compression.

The concepts of data compression are very well understood from books [12–15]. The field of data compression [14–19] has developed some of the most efficient and sophisticated compression algorithms. The compression process is complex and completed in two steps called decorrelation and entropy coding [20]. Decorrelation removes the inter-pixel redundancy by decorrelation techniques such as run-length coding, predictive techniques, transform techniques, or SCAN language-based methodology. The second step, entropy coding, removes coding redundancy. Entropy means how many average bits are required to represent a symbol. In coding, for frequently used symbols, fewer bits (value is less than entropy) are assigned, and more bits (value is more than entropy) are set to rarely used symbols. This leads to the formation of VLCs (Variable Length Codes). There are multiple VLCs proposed called unary codes, binary codes, gamma codes, omega codes,

etc. However, the most famous are Huffman codes [20] and arithmetic codes [21]. The later concept of tokenization is used in new compression algorithms called LZ77 and LZ78 [22] (inspired by researchers Lempel and Ziv in 1977 and 1978). LZ77 and LZ78 are forms of dictionary-based coding. They are used for lossless compression. A dictionary with available codes will be made available to the system. A unique index value is provided to each code. These indices are provided when an existing code is encountered. Fresh entry will be made when new codes are encountered. The main advantage given by dictionary vase coding is adaptiveness. Moreover, they are faster and since these methods are not based on statistics, and there is no dependency of the quality of the model on the distortion of data. Multiple variants of these algorithms were proposed and used in JPEG and MPEGs. These algorithms use traditional pixel-wise distortion measurements; they are PSNR (Peak Signal-to-Noise Ratio) and MS-SSIM (Multiscale Structural Similarity) [23]. A new compression algorithm called Burrows–Wheeler transform makes a cluster of similar symbols and does compression. This method is currently being used in the Linux operating system and many network protocols in the TCP/IP stack. After that, dynamic statistical encoding is used, which is adaptable to the input given to the data compression algorithm. This kind of input will decide the entropy value. The value may be different for multimedia data and textual data.

*Applications of Video Compression*

Video compression is a desperate requirement for many real-time applications shown in Figure 2. In this era, the service providers have good bandwidth and are very cheap in cost. Thus, most of the crowd has started using the internet. It results in generating a tremendous amount of data. Most of the data generated are videos. Therefore, it is challenging to save all that data in a limited space. This section discusses various applications where efficient video compression techniques can be used in the 21st century.

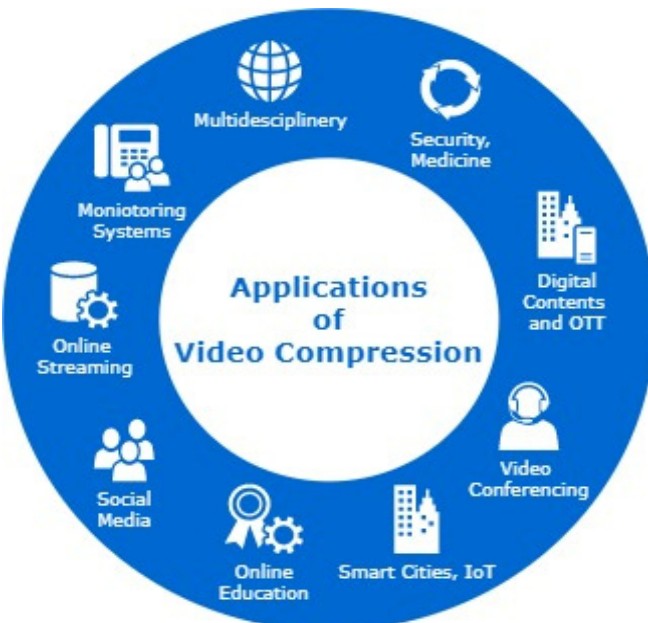

**Figure 2.** Applications of video compression.

- Video Conferencing: The cloud-based video conferencing platforms such as Microsoft Teams, Zoom, etc., kept educational systems and industries functional in this pandemic. Live sessions or meetings were ongoing continuously for 24 h of the day. High-quality live streaming of data is being transferred through the network. Our network is often not capable of transferring data of original quality to the receiver. Thus, efficient video compression technologies can help us achieve high-quality audio/video transfer through

the internet. Moreover, it will help in making the system cross-compatible in a heterogeneous hardware environment. A few approaches were proposed [24–26] for compression in video conferencing, but they have their own set of advantages and disadvantages.

- Social Media, Online education platforms, and OTT platforms: Today's generation spends lots of time on social media. Instagram, Facebook, LinkedIn, YouTube, and WhatsApp are the most widely used social media accounts. General and technical networking, sharing photos and videos, sharing achievement, and funny and entertainment content attract users to use them.

As per a survey by Finances online, around 4.2 billion users are for social media present today, and it will grow to 4.75 billion in the next half-decade [27]. Instagram/Facebook Reels, Stories, YouTube Shorts, etc., generate a humongous amount of video data. Moreover, OTT platforms such as Netflix, Hotstar, Amazon Prime, etc., have a huge amount of video data. In India, significant growth is seen in the subscribers and have a vast amount of video data available with them [28]. Moreover, platforms such as Coursera, Udemy, NPTEL, etc., have an enormous amount of video data. Service providers are required to deliver high-quality data to subscribers, especially OTT platforms and educational services. Although the proposed approaches provide satisfactory results [29–32], we need an efficient video compression algorithm for storing data and data streaming. Did you know that it requires approximately 750 GB of space to store a video of 90 min long with 1080 pixels and RGB color space? Moreover, it is delivered in 1 GB to 2 GB to the end-user. In particular, platforms such as Netflix face real-time problems such as preventing visual artifacts and film grain noise [33]. Thus, a high performing codec is a required component that will be able to handle such issues:

- Surveillance Video Applications: Applications such as smart traffic monitoring systems, drowsiness detection [24], identifying suspicious activities, CCTV [25], etc., can also require a high-quality codec to save data as well as retrieve data from the storage [26]. Maintaining data quality, object detection, object recognition, and semantically preserving objects or activities from videos is essential in such applications so that it will be a challenge to video codec.
- Multidisciplinary Applications: Currently, DL approaches are widely being used in the field of medicines, astronomy [34], security [35], autonomous driving cars [24], IoT [29–32], etc. In medicine [36,37], various surgeries are being recorded for records, educational purposes, or for future use. Moreover, videos recorded from space are stored for study purposes or may be used in applications relying on location-based services. The number of smart cities is growing. In smart cities, various IoT devices are capturing videos continuously for various purposes. As per a survey by Cisco, there will be around 22 billion cameras in the world by 2022. Storing and processing data in each application mentioned above is very challenging. An efficient codec may fulfill the requirement.

This paper shows a bibliometric study of all articles related to video compression. The analysis has been performed of all related publications and citations from the year 2018 to 2021 to outline the progress of work in the field of video compression. The analysis of work starts with published documents and author keywords. Then, the analysis comments on key organizations working in the area and their events. Then, it identifies potential journals that are contributing to the field's growth. Moreover, a later section in this paper will provide information about region-wise authors who are contributing to the area. The insights of this analysis are as follows:

1. To conduct a bibliometric study of the various video codecs used for compression;
2. To survey various deep learning-based approaches used by codecs for video compression;
3. To study various performance metrics and datasets available for the study of video compression;
4. To identify various real-time challenges to video codecs and what are future directions to the research.

Figure 3 shows organization of the paper. In this paper: the second section "Research Strategy" explains study techniques used for data collection and extraction and provides data analysis based on it. The third section "Quantitative Analysis" talks about the results of bibliometric analysis with graphical pictures drawn using software named "VOS Viewer" and "Gephi". The fourth section "Qualitative Analysis" outlines video compression and research trends available for it. The paper concludes with a section on "Discussion", which highlights significant challenges current approaches are facing, the scope to improve the work, important findings from the analysis, and future directions in the field.

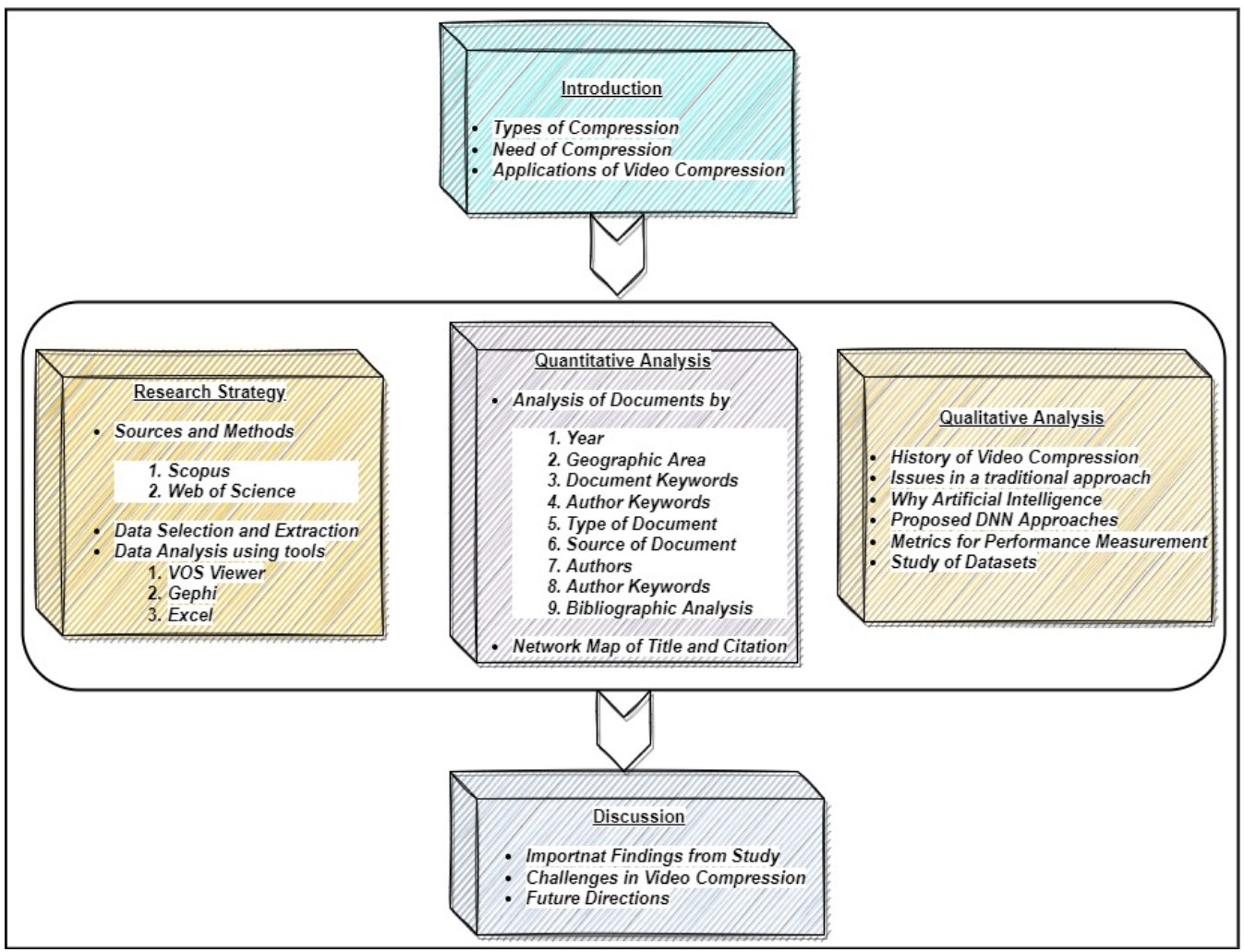

**Figure 3.** Organization of paper.

## 2. Research Strategy

### 2.1. Source and Methods

Bibliometrics is one of the frequently used terms in research evaluation metrics. Bibliometrics is a set of methods for quantitatively analyzing academic literature and scholarly communications [38,39]. In bibliometric analysis, we rigorously explored and analyzed extensive scientific data. This analysis helps us create high research impact, obtain a one-stop overview, identify knowledge gaps, and derive novel ideas that can be investigated further [40]. In bibliometric analysis, published research is accessed based on fundamental metrics. In this process, we try to mine the most active or prominent researchers and their organizations, collaboration patterns, frequently used keywords, and various articles published on them. All this required information can be made available from famous repositories called "Scopus" and "Web of Science". Scopus [41] is a well-known and largest

peer-reviewed abstract and citation database introduced by Elsevier in 2004. Web of Science [42] is owned by Thomson Reuter, containing abstract and citation databases of SCI and SSCI publications.

### 2.2. Data Selection and Extraction

The following table represents the search strategy used for finding results on the Scopus and Web of Science databases. Adding appropriate and relevant keywords in the search bar while finding literature is very important. The keywords to find relevant documents are chosen after studying the latest survey papers [43,44] of video compression.

The fundamental keyword used for the search is "Video Compression", and two different keywords found into abstracts are "Video Compression" and "Neural network". Various neural network algorithms are used for compression, so an umbrella term called "Neural Networks" is provided for the search instead of providing a separate algorithm name. "Compression" is the only keyword assigned to query from the keyword section from research papers.

The detailed query is given below in the table. All results up until 2021 are considered from the total generated results. Figure 4 shows relevant research papers found after giving queries to databases. Only journals, conferences, and review papers are considered for the analysis. The query has resulted in only 84 articles from the Scopus database, and 36 pieces were extracted from the Web of Science database.

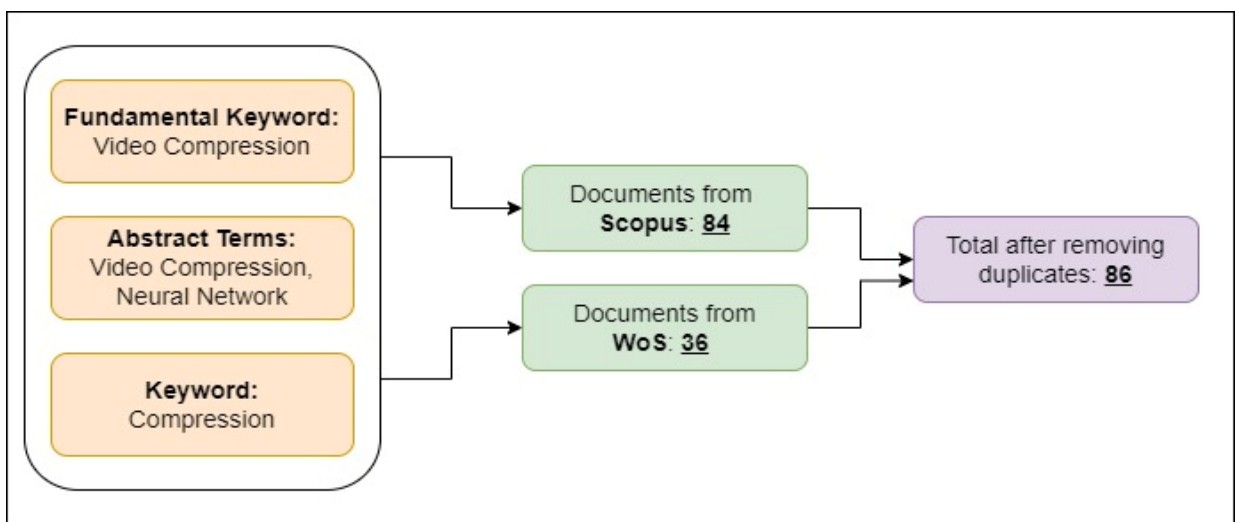

**Figure 4.** Search Strategy.

After removing duplicates, only 86 documents were found. For the retrieved research papers, metadata is extracted, containing the paper title, publication year, source, a number of citations, author's name, author's keywords, cited references, organization, and country. The results used for the analysis were retrieved on 4 January 2022. Table 1 represents the fundamental keywords used in the search strategy.

**Table 1.** List of keywords used in the table.

| Fundamental Keyword | Video Compression |
|---|---|
| Primary Keyword Using "AND" | Neural Networks |
| Secondary Keywords Using "OR" | "GAN", "Generative Adversarial Network", "CNN", "Convolutional Neural Network" |
| Author's Keywords Using "OR" | "Video Compression", "Compression" |

- Query in Scopus:

(TITLE (video AND compression) AND ABS (video AND compression) AND ABS (neural AND networks) AND KEY (compression) OR ABS (gan) OR ABS (autoencoders) OR ABS (generative AND adversarial AND network) OR ABS (CNN) OR ABS (convolutional AND neural AND network)).

- Query in Web of Science

("Video Compression" (Title), AND "Neural Networks" (Abstract) AND "Video Compression" (Abstract) AND "Video Compression" (Author's Keyword) OR "Compression" (Author's Keyword)).

### 2.3. Data Analysis Procedure

Information can be easily understood and analyzed if it is represented in graphs. It also helps in concluding, decision making, predictions, etc. In this paper, bibliometric analysis of deep neural networks techniques used for video compression is completed using software "VoSViewer" [45], "Gephi" [46], and "BibExel" [47]. These comprise software prevalent for the representation of multidimensional data in graphical visualization. VoSViewer is a very popular visualization tool for bibliometric analysis. We can make various networks based on the keywords, citations, source of publishing, authors, co-citation, etc. Circles represent all respective objects. They are mapped to other objects through links. The distance between different objects represents the association between them. The smaller the distance, the more closely associated objects are. Gephi is a very popular graphical clustering tool. It is a cross-platform software that uses OpenGL 3D engine. It allows us to configure data according to its scale, properties, classification, etc. BibExel is a tool developed by Olle Persson, an information scientist. BibExel is a free software designed for non-profit educational use. It is another tool used to assist researchers with bibliometric analysis.

Analysis completed in this paper is divided into two major categories: quantitative and qualitative. As mentioned earlier, the details are extracted from two famous databases: Scopus and Web of Science. The following analysis are completed in quantitative analysis:

- Analysis of documents by year;
- Citation based analysis;
- Top keywords from Scopus and Web of Science;
- Analysis of document type;
- Analysis by geographical area;
- Analysis of publication by source;
- Co-Occurrence analysis (author keywords).

Qualitative analysis will focus on video compression, its history, deep learning-based approaches proposed, performance metrics, and datasets available for study. The latest proposed techniques have used famous DNN approaches, including CNN, GAN, Autoencoders, etc.

## 3. Quantitative Analysis

### 3.1. Analysis of Documents by Year

The research in video compression started in the late 1990s. Traditional research mainly focused on interframe or intraframe-based compression, which uses pairs of electronic circuits such as quantizers–dequantizer, transforms–reverse transformers, etc. However, since the contribution of video data generated in big data increased lately, it has been researched extensively. Interdisciplinary approaches are helping to explore various possibilities of video compression. Recently, deep learning-based approaches are widely used. Details of these approaches will be learned in qualitative analysis. Figure 5 shows the year-wise analysis of the documents published. The number of papers published in the field has increased since 2018.

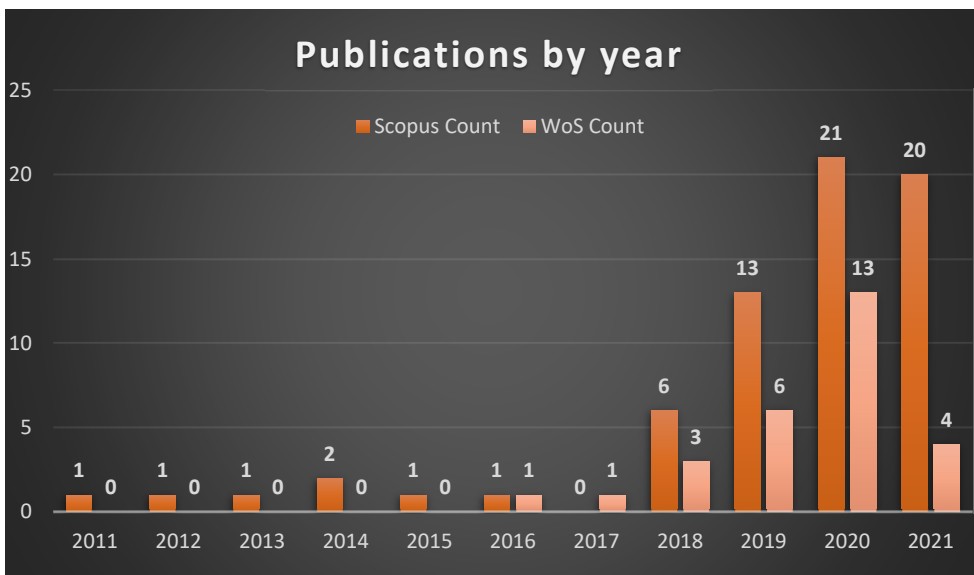

**Figure 5.** Comparative analysis of publications per year.

### 3.2. Citation Based Analysis

A number of citations in the published document explained the need and the relative significance of the solution to the problem provided by the document. Table 2 shows an analysis of year-wise citations received for documents published in Scopus and WoS databases. The citation count has been increasing since 2019, which shows that a significant amount of work is being carried out currently in the world.

**Table 2.** Year-wise citation analysis.

| Year | <2017 | 2017 | 2018 | 2019 | 2020 | 2021 | Total |
|---|---|---|---|---|---|---|---|
| Scopus Citation | 125 | 8 | 13 | 35 | 151 | 210 | 542 |
| Web of Science Citation | 18 | 1 | 3 | 21 | 57 | 88 | 188 |

The top five publications from the Scopus database are given in Tables 3 and 4 and provides information on the top five publications from the WoS database. Publications are ranked based on their citation count. Figure 6 is an Alluvial diagram showing an analysis of the top 20 cited publications in the field of study from the Scopus database. It provides a correlation between authors, year of publication, source, and citation count of the highly cited documents.

**Table 3.** Top 5 publication (as per citations) in Scopus.

| References and Years | Authors | <2017 | 2017 | 2018 | 2019 | 2020 | 2021 | Total |
|---|---|---|---|---|---|---|---|---|
| [48] (2021) | Lu G. et al. | 0 | 0 | 0 | 4 | 39 | 34 | 77 |
| [49] (1996) | Gelenbe E. st al. | 50 | 2 | 8 | 0 | 2 | 3 | 65 |
| [44] (2020) | Ma S. at al. | 0 | 0 | 0 | 2 | 11 | 27 | 40 |
| [50] (2018) | Chen T. et al. | 0 | 0 | 1 | 9 | 16 | 14 | 40 |
| [51] (2019) | Djelouah A. et al. | 0 | 0 | 0 | 1 | 17 | 13 | 31 |

**Table 4.** Top 5 publication (as per citations) in WoS.

| References and Years | Authors | <2017 | 2017 | 2018 | 2019 | 2020 | 2021 | Total |
|---|---|---|---|---|---|---|---|---|
| [44] (2020) | Ma. S. et al. | 0 | 0 | 0 | 5 | 12 | 27 | 44 |
| [52] (2019) | Afonso, Mariana, et al. | 0 | 0 | 0 | 8 | 9 | 7 | 24 |
| [50] (2018) | Chen T. et al. | 0 | 0 | 1 | 4 | 9 | 8 | 22 |
| [53] (2019) | Kaplanyan, Anton et al. | 0 | 0 | 0 | 0 | 11 | 10 | 21 |
| [54] (1998) | Cramer C. | 9 | 0 | 1 | 0 | 1 | 0 | 11 |

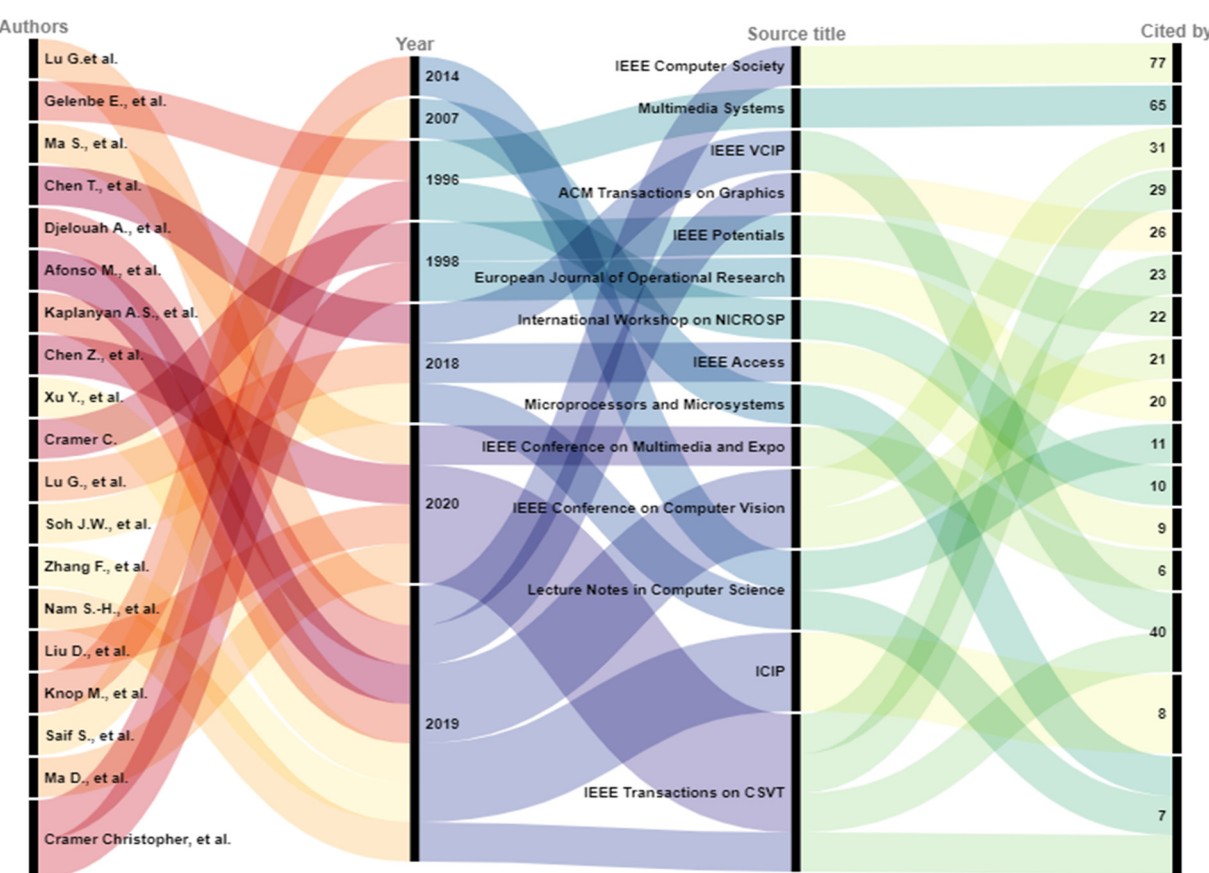

**Figure 6.** Alluvial diagram showing a correlation between authors, years, and source titles of top 20 cited documents.

## 4. Research Virtue

### 4.1. Top 10 Keywords from Scopus

Figure 7 shows a treemap of the top 10 author keywords from documents extracted from the Scopus database. Image compression is observed to occur the most, numbering 67. Video compression is ranked second if we arrange them in decreasing order. We will analyze author keywords in co-occurrence analysis. The analysis is performed on all available documents in the domain.

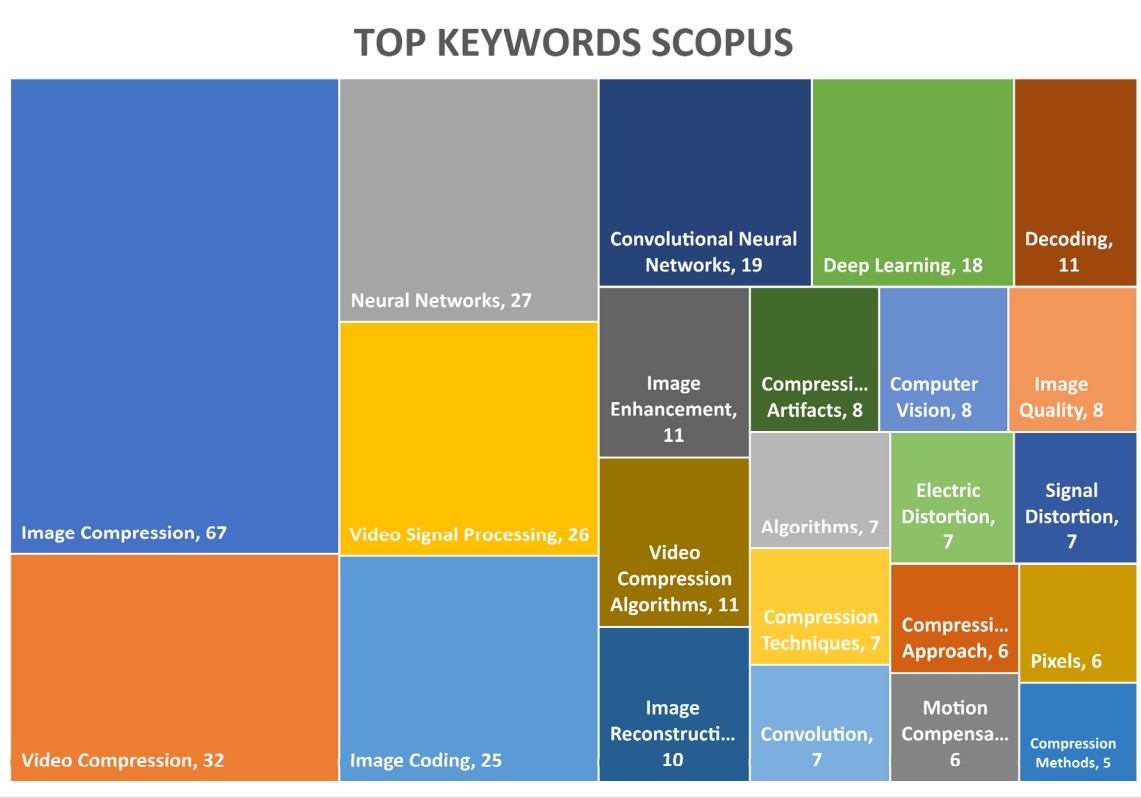

**Figure 7.** Top keywords used in Scopus.

### 4.2. *Analysis of Document Type*

Table 5 shows details of documents published in the field of video compression. A total of 121 papers are published in Scopus indexed and WoS indexed events. The detailed distribution of publishing categories is visualized in Figure 8.

It has been observed from Figure 8 that more than 50% of documents are published in conferences. The last major category with respect to publishing category is journal/articles. Very few survey papers and only book chapters are published in the domain. However, no bibliometric analysis is found in the field of video compression. A detailed analysis of sources of the publication and their citations count is discussed in one of the later sections.

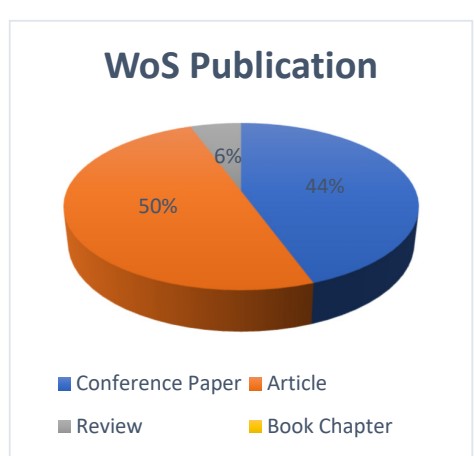
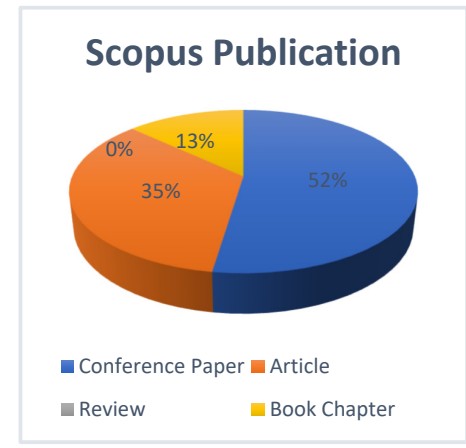

**Figure 8.** Category of publication.

**Table 5.** Publication count by document type.

| Type of Publication | Scopus | Web of Science | Total |
|---|---|---|---|
| Conference Paper | 44 | 16 | 60 |
| Article/Journal | 29 | 18 | 47 |
| Review | 00 | 02 | 02 |
| Book Chapter | 11 | 00 | 11 |
| | **84** | **37** | **121** |

*4.3. Analysis of Geographical Area*

Analysis of documents published region-wise or country-wise gives information about ongoing research in respective countries. Figures 9 and 10 provide information about the country-wise count of documents published worldwide in Scopus and WoS indexed publications.

In Scopus indexed publications, China has the topmost publication count, which is 21, and USA comes after. India is third on the list, which explains that substantial quality research is ongoing in India in video compression.

In WoS indexed publications, Russia comes first. The National Science Foundation (NSF); Google; and National Natural Science Foundation of China are the major contributing funding agencies to the research in the domain. Howard Hughes Medical Institute, British Broadcasting Corporation (BBC); Engineering and Physical Sciences Research Council (EPSRC); Ministry of Education—Singapore; Ministry for Science; ICT of the Korean Government; and Nvidia are other sources who are helping research. University Grants Commission (UGC) has funded ongoing research in India. Tables 6 and 7 provide the top seven countries with publication counts. The analysis is performed on all available documents in the domain.

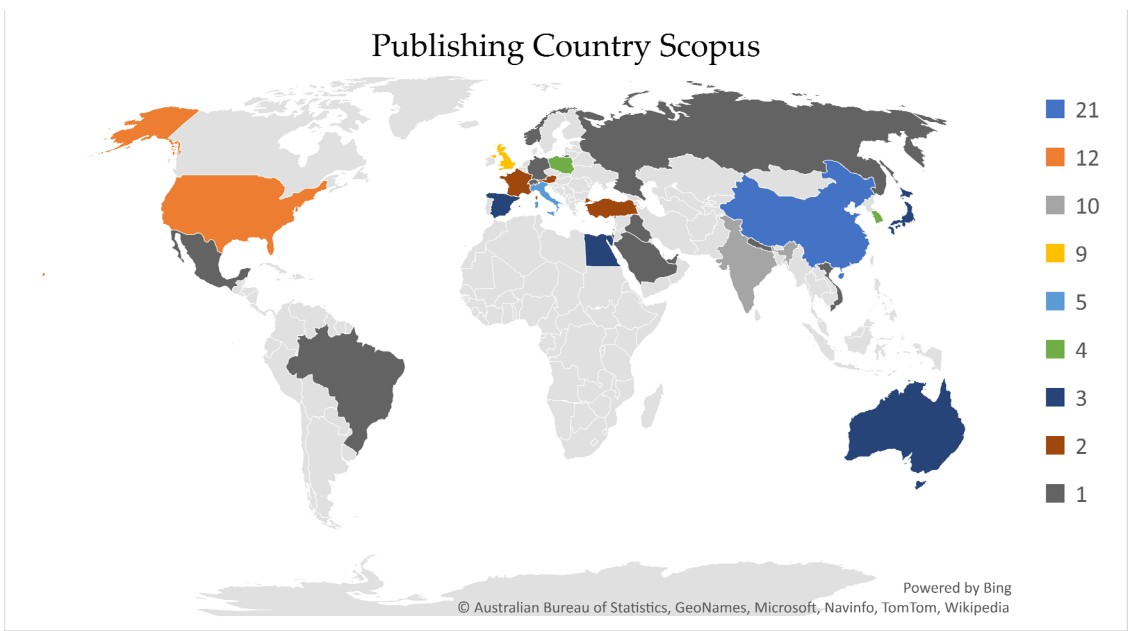

**Figure 9.** Publishing country: Scopus.

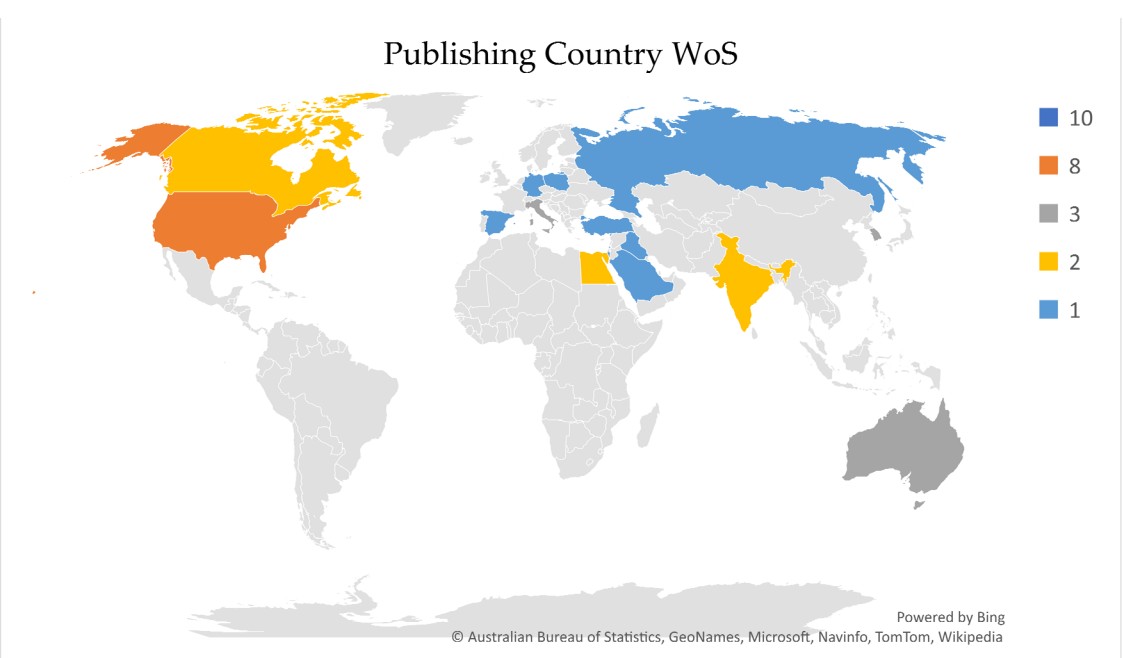

**Figure 10.** Publication country: WoS.

**Table 6.** Country-wise publication count WoS.

| Country | Count |
| --- | --- |
| China | 10 |
| USA | 8 |
| Australia | 3 |
| Italy | 3 |
| South Korea | 3 |
| Egypt | 2 |
| India | 2 |

**Table 7.** Country-wise publication count Scopus.

| Country | Count |
| --- | --- |
| China | 21 |
| USA | 12 |
| India | 10 |
| UK | 9 |
| Italy | 5 |
| Poland | 4 |
| Australia | 3 |

*4.4. Analysis of Publication by Source*

Figures 11 and 12 show top sources published documents in Scopus and WoS indexed publications. IEEE is a primary source where most of the papers are published. Springer Nature is the next source where most researchers have published their work. Computer Vision Foundation (CVF) is an organization that mainly publishes research in image and video compression. The analysis is performed on all available documents in the domain.

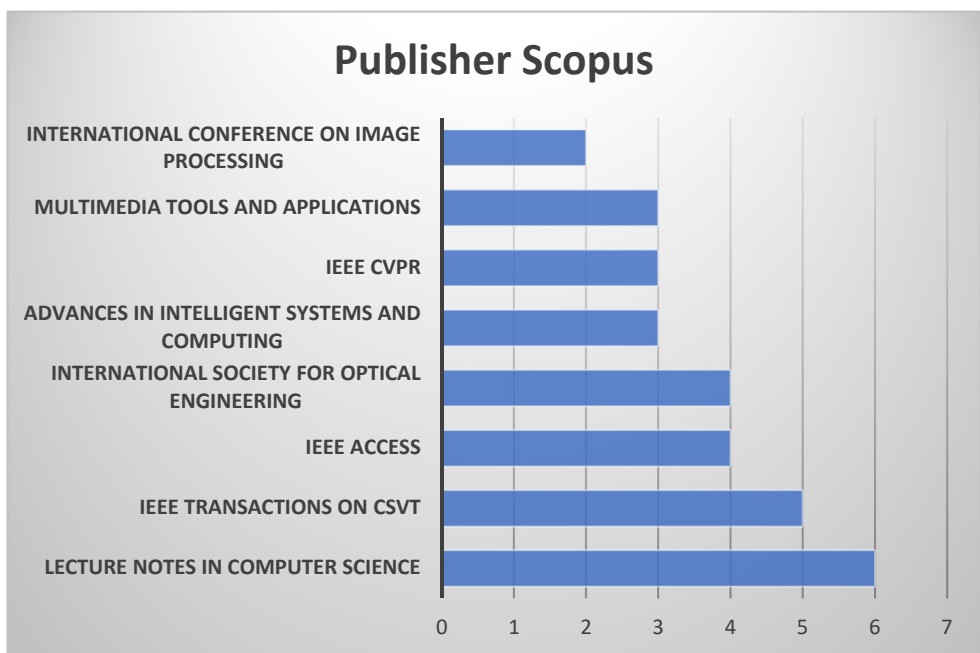

**Figure 11.** Publishers in Scopus.

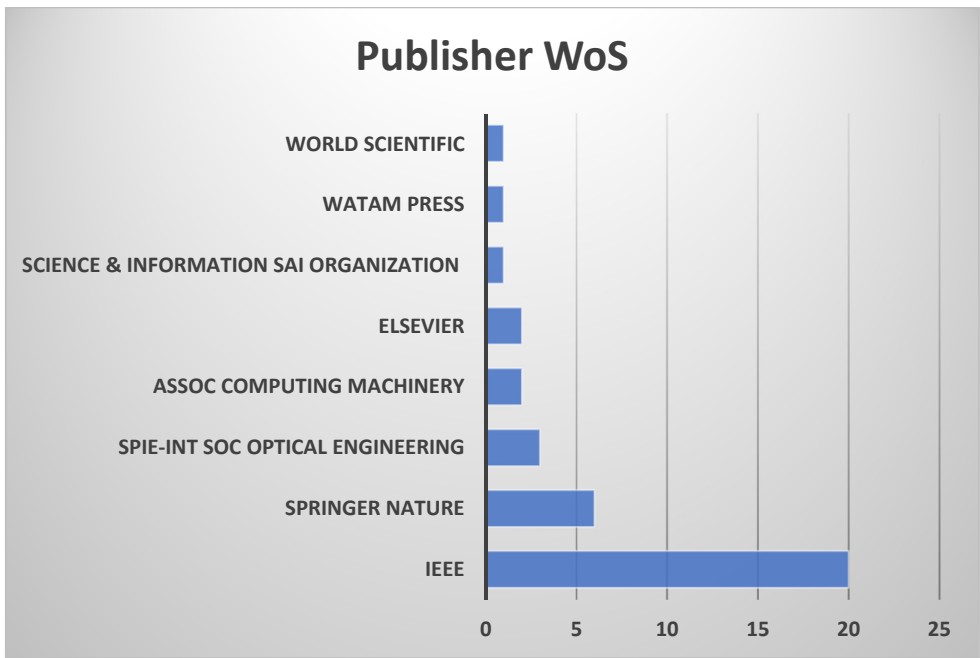

**Figure 12.** Publishers in WoS.

*4.5. Co-Occurrence Analysis (Author Keywords)*

Figure 13 shows the co-occurrence analysis of author keywords extracted from unique documents exported from Scopus and WoS databases. Video compression is the keyword used the most significant number of times. Other keywords mainly used are Deep Learning, CNN, High-Efficiency Video Coding (HEVC), etc.

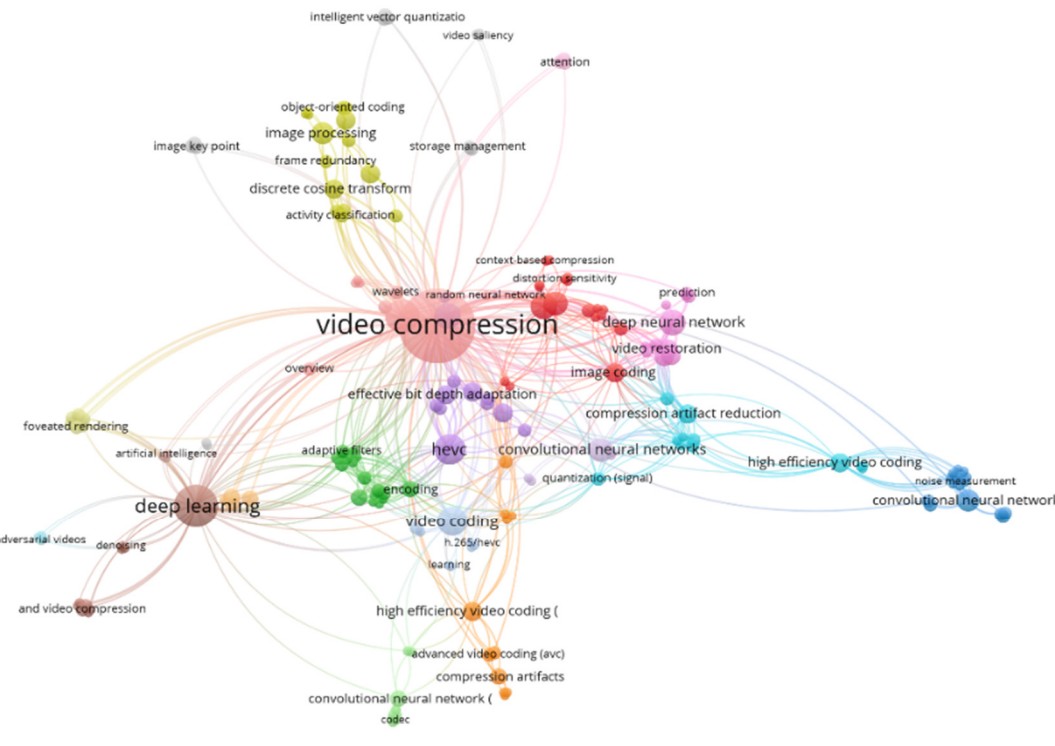

**Figure 13.** Co-occurrence analysis (author keywords).

Table 8 provides information about keywords, the number of links of that keyword, and its Total Link Strength (TLS) value. Link strength and TLS are weighted attributes. Link attribute is a measure of co-authorship of a given researcher with other researchers, and TLS represents the total strength of co-authorship links between the respective author and other researchers.

**Table 8.** Country-wise publication count Scopus.

| Keyword | Occurrence | Number of Links | Total Link Strength (TLS) |
|---|---|---|---|
| Video Compression | 40 | 100 | 144 |
| Deep Learning | 15 | 46 | 64 |
| Convolutional Neural Network/s (CNN) | 11 | 45 | 45 |
| Neural Network/s | 11 | 29 | 42 |
| High-Efficiency Video Coding(HEVC) | 10 | 42 | 49 |
| Video Coding | 7 | 26 | 34 |
| Image Compression | 6 | 17 | 25 |
| Deep Neural Network | 4 | 9 | 13 |
| Rate distortion optimization | 3 | 12 | 16 |
| Image Processing | 3 | 9 | 10 |
| Image Coding | 3 | 9 | 10 |
| Cellular Neural Networks | 3 | 6 | 8 |
| Image/Video Compression | 3 | 17 | 7 |
| Encoding | 2 | 17 | 18 |
| Transform coding | 2 | 15 | 17 |
| HD video | 2 | 10 | 13 |
| Spatiotemporal Saliency | 2 | 10 | 13 |
| Compression Artifact reduction | 2 | 8 | 8 |
| Discrete Cosine Transform | 2 | 7 | 8 |
| Effective bit depth adaptation | 2 | 7 | 8 |

The keyword video compression has the highest TLS value, i.e., 144. The keywords Deep Learning, CNN, and Neural Network/s have a combined TLS value of 151. This value shows that many approaches to video compression using advanced ML techniques are being examined. The analysis is performed on all available documents in the domain.

### 4.6. Citation Analysis of Documents

The paper's citation count shows the impact of the work in the form in that domain. Co-citation analysis will result in finding the influential publication. Figure 14 and Table 9 show a detailed analysis of citations of documents.

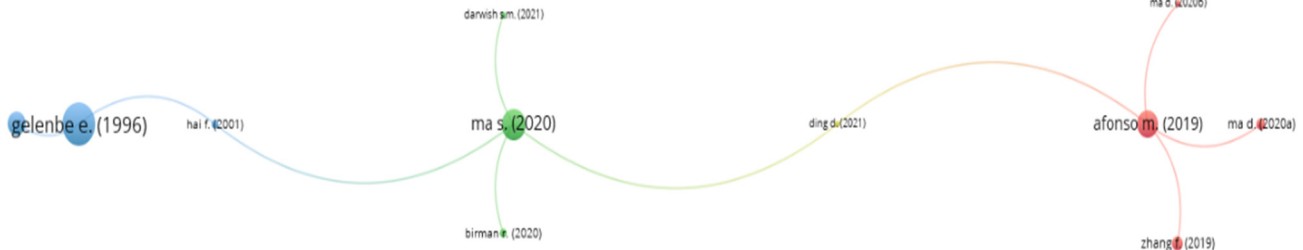

**Figure 14.** Citation analysis of documents.

**Table 9.** Top 12 documents with highest citations.

| Document Author | Citations | Links |
| --- | --- | --- |
| Lu G. (2019) | 75 | 0 |
| Gelenbe E. (1996) | 65 | 2 |
| Ma S. (2020) | 37 | 4 |
| Chen T. (2018) | 37 | 0 |
| Djelouah A. (2019) | 27 | 4 |
| Afonso M. (2019) | 27 | 4 |
| Kaplanyan A.S. (2019) | 22 | 0 |
| Cramer Christopher (1998) | 22 | 0 |
| Chen Z. (2020) | 21 | 2 |
| Cramer C. (1998) | 20 | 1 |
| Xu Y. (2019) | 18 | 0 |
| Lu G. (2018) | 11 | 0 |

All research papers from both databases are considered. Lu g. (2019) has the highest number of citations, i.e., 75. The number of documents and their citation count explains that much work needs to be conducted in video compression. The analysis is performed on all available documents in the domain.

### 4.7. Citation Analysis of Source

Figure 15 and Table 10 provide detailed information of sources where a paper in the field of study is published. As per the analysis performed on document type, around 50% of articles are published in conferences. Therefore, most of the papers are published in proceedings of the conferences. IEEE transactions had published the most notable number of documents in the domain. It has seven articles published in it. "*Lecture Notes in Computer Science*" is the next source published by six papers. Conferences associated with IEEE, IEEE potentials, and IEEE access are favorite sources. The analysis is performed on all available documents in the domain.

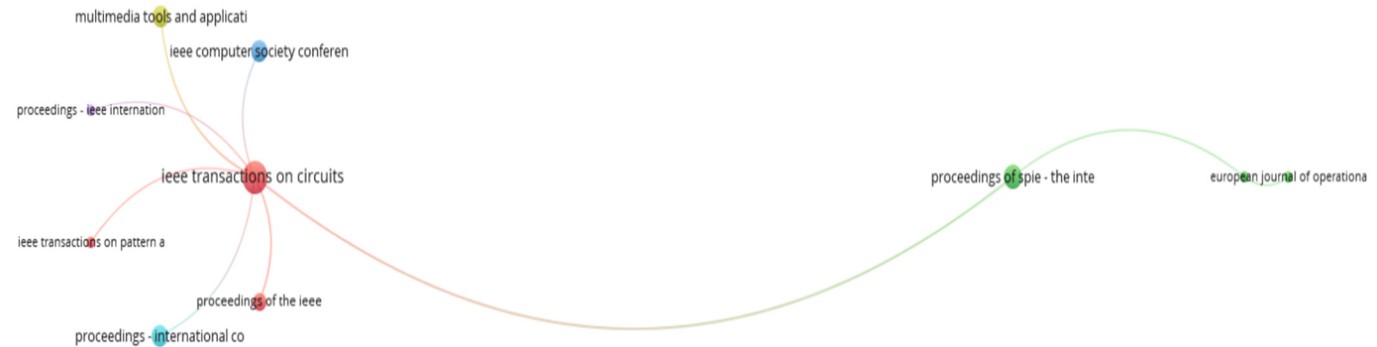

**Figure 15.** Citation analysis of documents.

**Table 10.** Citation analysis by source.

| Source | Documents | Citations | Links | TLS |
|---|---|---|---|---|
| IEEE transactions on circuits and systems for video technology | 7 | 102 | 7 | 10 |
| Lecture Notes in Computer Science | 6 | 23 | 1 | 0 |
| IEEE access | 4 | 9 | 1 | 0 |
| International conference on image processing, ICIP | 3 | 17 | 1 | 1 |
| IEEE international conference on computer vision | 2 | 47 | | 0 |
| IEEE computer society conference on computer vision and pattern recognition | 1 | 75 | 1 | 0 |
| Multimedia systems | 1 | 65 | 2 | 2 |
| IEEE visual communications and image processing, VCIP 2017 | 1 | 37 | 1 | 0 |
| ACM transactions on graphics | 1 | 22 | 1 | 0 |
| IEEE potentials | 1 | 22 | 1 | 0 |
| European journal of operational research | 1 | 20 | 1 | 1 |
| International workshop on neural networks for identification, control, robotics, and signal/image processing, NICROSP | 1 | 8 | 1 | 0 |

*4.8. Citation Analysis of Author*

Zhang X. is the author who has published the highest number of documents, i.e., 5, and has the most cumulative citations, i.e., 131. Gao z., Lu g., Ouyang w., Xu d., Bull d.r., and Zhang f. have published four documents each. Detailed analysis is provided in Figure 16 and Table 11. The study was performed on all available documents in the domain.

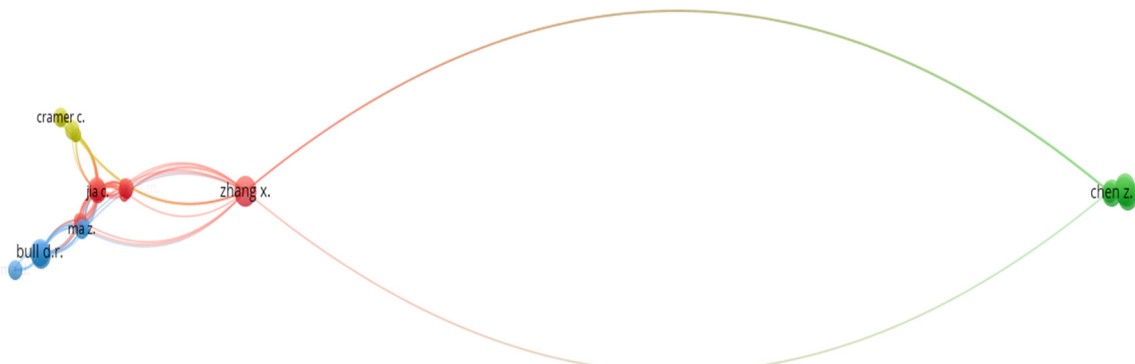

**Figure 16.** Citation analysis by author.

**Table 11.** Citation analysis by author.

| Name of Author | Documents | Citations | Links | TLS |
|---|---|---|---|---|
| Zhang X. | 5 | 131 | 19 | 19 |
| Gao Z. | 4 | 94 | 4 | 4 |
| Lu G. | 4 | 94 | 4 | 4 |
| Ouyang W. | 4 | 94 | 4 | 4 |
| Xu D. | 4 | 94 | 4 | 4 |
| Bull D.R. | 4 | 40 | 9 | 18 |
| Zhang F. | 4 | 40 | 9 | 18 |
| Cramer C. | 2 | 85 | 6 | 7 |
| gelenbe E. | 2 | 68 | 11 | 12 |
| Cai C. | 1 | 75 | 0 | 0 |
| Gelenbe P. | 1 | 65 | 5 | 5 |
| Sungur M. | 1 | 65 | 5 | 5 |

*4.9. Bibliographic Coupling of Documents*

Bibliographic coupling explains that if two documents share references, it also has the same technical content. Figure 17 and Table 12 provide a detailed analysis of the bibliographic coupling of all documents. Lu.g. is the author with the highest TLS value 109, links 39. The study is performed on all available documents in the domain.

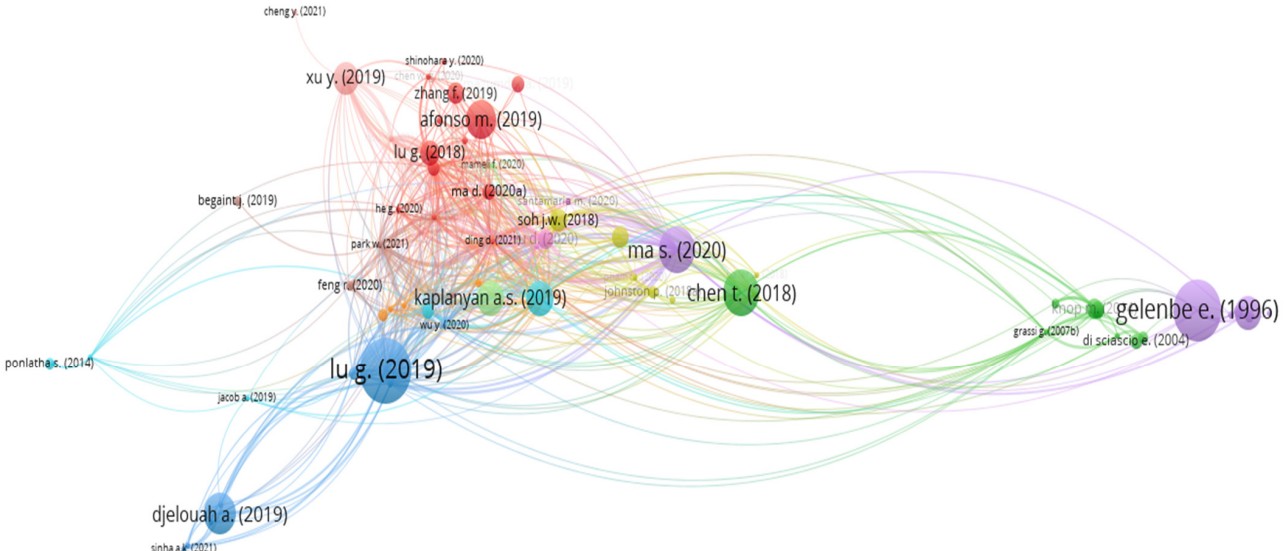

**Figure 17.** Bibliographic analysis of documents.

**Table 12.** Bibliographic analysis of documents.

| Document | Citations | Links | TLS |
|---|---|---|---|
| Lu G. (2019) | 75 | 39 | 109 |
| Gelenbe E. (1996) | 65 | 9 | 18 |
| Ma S. (2020) | 3 | 40 | 156 |
| Chen T. (2018) | 37 | 27 | 36 |
| Djelouah A. (2019) | 29 | 10 | 26 |
| Afonso M. (2019) | 27 | 22 | 31 |
| Kaplanyan A.S. (2019) | 22 | 15 | 22 |
| Cramer Christopher (1998) | 42 | 6 | 18 |
| Chen Z. (2020) | 21 | 38 | 108 |
| Xu Y. (2019) | 18 | 27 | 70 |
| U G. (2018) | 11 | 33 | 97 |
| Soh J.W. (2018) | 9 | 33 | 111 |

*4.10. Network Map of Publication Title and Citation*

Figure 18 is a network map of titles of the publication with citations. The network metric enriches the analysis of the documents. This kind of analysis focuses on authors' relative importance, institutions, country, etc., by using network metrics such as PageRank, eigenvector centrality, degree of centrality, betweenness centrality, and closeness centrality.

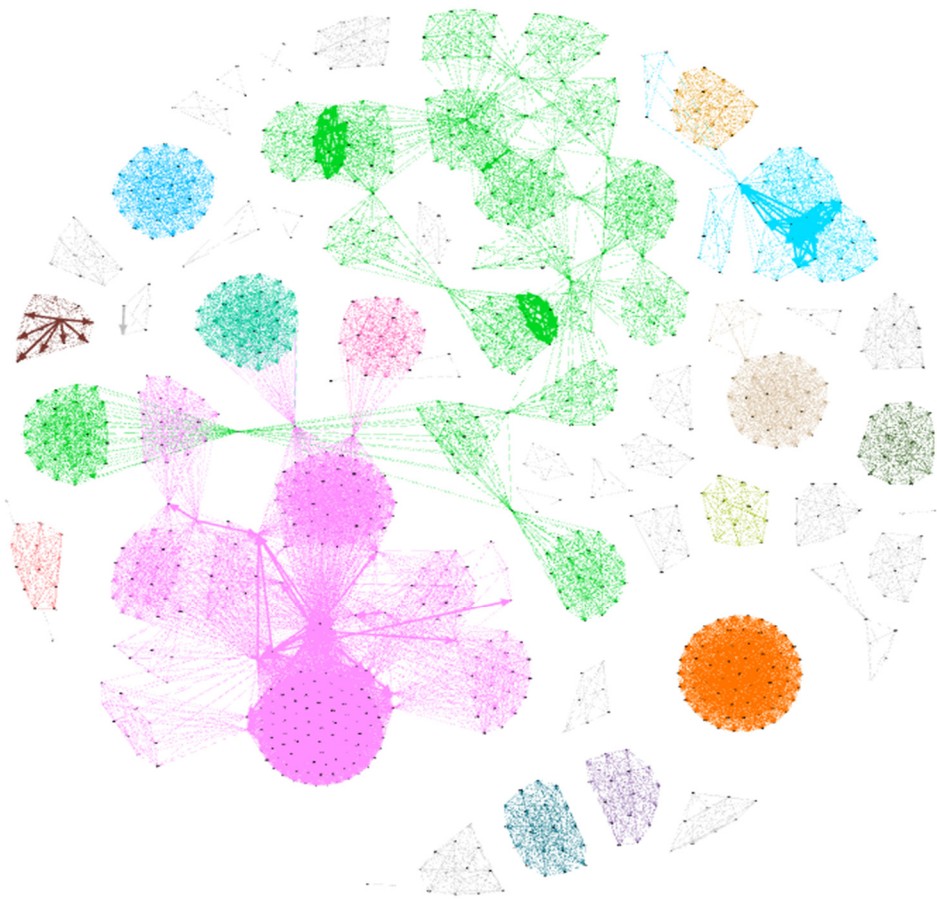

**Figure 18.** Title of the publication and citations network visualization.

PageRank represents the impact of a document. The same method is used for prioritizing web pages. The degree of centrality explains how many papers are published by a researcher as an author or coauthor in a particular domain. If the value is five, they must be an author or co-author for five documents. How much two related documents carry out information is measured by betweenness centrality. Eigenvector centrality measures how many highly valued documents relate to a particular document. The higher the value, the more importance is attached to that document. Closeness centrality means how a document is closely related to other important documents in the network The network map is drawn using Gephi's Fruchterman Reingold layout. The network map consists of clusters with different colors; each cluster provides information on publications sharing similar citations. There are a total of 999 nodes and 16,349 connected edges. Following tables provides information of top 5 documents of respective metrics: Table 13 (PageRank), Table 14 (Degree of Centrality), Table 15 (Betweenness Centrality), Table 16 (Eigen Centrality) and Table 17 (Closeness Centrality).

**Table 13.** Top 5 documents with highest PageRank.

| Title | PageRank |
|---|---|
| **Overview of the High-Efficiency Video Coding (HEVC) Standard (2012)** | 0.003829 |
| **Adam: A Method for Stochastic Optimization (2014)** | 0.003235 |
| **Image Quality Assessment: From Error Visibility to Structural Similarity (2004)** | 0.002478 |
| **HEVC Deblocking Filter (2012)** | 0.002395 |
| **Sample Adaptive Offset in the HEVC Standard (2012)** | 0.002395 |

**Table 14.** Top 5 documents with the highest degree of centrality.

| Title | Eccentricity |
|---|---|
| **Convolutional LSTM Network: A Machine Learning Approach for Precipitation Nowcasting (2015)** | 7 |
| **Iterative Procedures for Reduction of Blocking Effects in Transform Image Coding (1992)** | 7 |
| **Characterizing Perceptual Artifacts in Compressed Video Streams (2014)** | 7 |
| **Multi-Frame Quality Enhancement for Compressed Video (2018)** | 7 |
| **Image Restoration by Estimating Frequency Distribution Of Local Patches (2018)** | 7 |

**Table 15.** Top 5 documents with highest betweenness centrality.

| Title | Betweenness Centrality |
|---|---|
| **Image Quality Assessment: From Error Visibility to Structural Similarity (2004)** | 13,624.71111 |
| **Overview of The High-Efficiency Video Coding (HEVC) Standard (2012)** | 12,780.45105 |
| **Compression Artifact Reduction by Overlapped-Block Transform Coefficient Estimation with Block Similarity (2013)** | 10,800 |
| **Adam: A Method For Stochastic Optimization (2014)** | 10,625.44351 |
| **Neural Network Approaches To Image Compression (1995)** | 8439 |

**Table 16.** Top 5 documents with highest Eigen centrality.

| Title | Eigen Centrality |
|---|---|
| **Overview of the High-Efficiency Video Coding (HEVC) Standard (2012)** | 1 |
| **HEVC Deblocking Filter (2012)** | 0.966484 |
| **Sample Adaptive Offset in The HEVC Standard (2012** | 0.966484 |
| **Adaptive Loop Filtering for Video Coding (2013)** | 0.955361 |
| **A Gaussian Denoiser: Residual Learning of Deep CNN for Image Denoising (2017)** | 0.914142 |

**Table 17.** Top 5 documents with highest closeness centrality.

| Title | Closeness Centrality |
|---|---|
| **A Statistical Evaluation of Recent Full Reference Image Quality Assessment Algorithms (2006)** | 1 |
| **Overview of the High-Efficiency Video Coding (HEVC) Standard (2012)** | 1 |
| **Convolutional LSTM Network: A Machine Learning Approach for Precipitation Nowcasting (2015)** | 1 |
| **Non-Local Structure-Based Filter for Video Coding (2015)** | 1 |
| **Interweaved Prediction for Video Coding (2020)** | 1 |

## 5. Qualitative Analysis

Almost everyone in the industry is using video compression several times a day. Including streaming a video on YouTube, Shorts or Reels on Instagram and Facebook, OTT platforms, online education, etc., all these rely on video compression technology heavily. The definition of video compression is to reduce the data used to encode digital video content.

It results in lower transmission bandwidths and lower memory requirements for the video. The codec used in video compression must take care that there should not be a significant degradation in the visual experience of the video content; also, it should not generate considerable hardware overhead in the process of compression. A video codec may be a software or an electronic subsystem that can cause compression or decompression of digital video.

It compresses raw or uncompressed video data to compressed format and vice versa. Video codec can be briefly divided into two parts: 'encoder', which performs compression, and 'decoder', which takes care of decompression. Video compression can be either lossy or lossless. The video codec provides various levels of compression. The aggressiveness of the compression is directly proportional to the savings in the storage space and the amount of bandwidth required for transmission. Increased aggressiveness degrades the quality of the video content (affects visual artifacts, blurriness, haze, etc.). Moreover, it requires extra hardware efforts and computing power to achieve that.

Thus, it is a more significant challenge to decide the level of compression we should perform. The other challenge current video compression is facing is to adapt to the latest video formats evolving in the world. We will discuss these challenges and their solutions later in the paper. The following paragraph provides a brief history of video compression from the beginning. Figure 19 showing a timeline of evolution of compression algorithms.

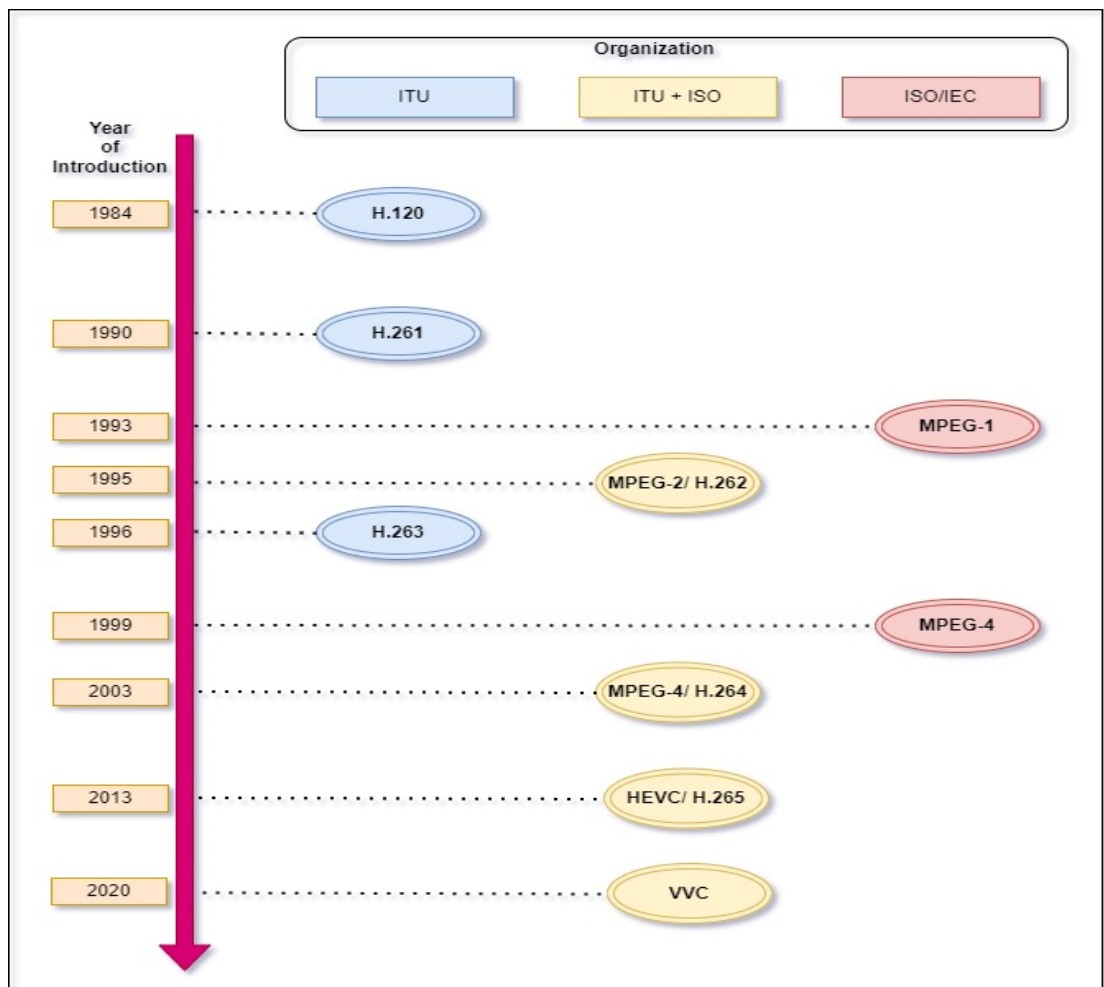

**Figure 19.** Timeline of video compression algorithms.

*5.1. History of Video Compression*

In 1929, Ray Davis Kell described a form of video compression and was also granted a patent for it. After that, many efficient video compression standards have been proposed, and they are still an integral part of today's video compression standards. The main challenge to video codecs is to represent video data more compactly and robustly where video transmission can lead to less cost in terms of bandwidth, power consumption, and, the most important factor, i.e., memory space. H.120 is the first digital video technology standard recommended by the International Telecommunication Union (ITU) in 1984. The objective behind proposing H.xxx standards by ITU was video conferencing. In 1990, ITU H.261 [55] was proposed, which was the first practical video compression approach.

The target of H.261 was to transmit videos over the communication line. In 1993, the International Standards Organization (ISO) and International Electrotechnical Commission (IEC) introduced the world to the famous MPEG family. The first compression algorithm proposed was MPEG-1 [56], widely used in video-CD. Later in 1995, MPEG-2 [57] was the compression algorithm used in DVDs, and it also could support HDTV (High-Definition Television). H.263 [58] was later proposed in 1996, which brought a revolution in video streaming and video conferencing applications. MPEG-4 [59], introduced in 1996, enables watching videos online. It used encoding technology called DivX, which has a crucial contribution in the pre-HD era. DivX uses AVI file extensions. XviD is an open-source version of DivX. It enables playing all videos that are using DivX files. Later in 2003, H.264 [60] was proposed, which is very popular for HD streaming of the data. It supports Blue-ray, HD DVD, digital video broadcasting, iPod video, Apple TV, and video conferencing. Later

in 2013, H.265 [61] was introduced from live HD broadcasting. All the above standards are already being available in the market, and they are trying to provide high-performing services for all (enterprises and customers). Moreover, they are adapted to the challenging real-time environment of an application of distance learning, live HD broadcasting, video conferencing, short video platforms, online gaming, e-commerce, etc. An ITU and MPEG committee has started developing a new standard called Versatile Video Coding (VVC) [62] to replace H.265. A comprehensive comparison of all the above algorithms can be further read from [63]. Table 18 is providing brief summary of characteristics of video codecs.

**Table 18.** Video codecs and their characteristics.

| Video Compression Algorithm | Family | Year of Introduction | Characteristics |
|---|---|---|---|
| H.120 | H.xxx | 1984 | First standard by ITU. Used for video conferencing |
| H.261 | H.xxx | 1990 | First practical video compression approach. They are used for video transmission over the communication line. |
| MPEG-1 | MPEG | 1993 | First compression algorithm by MPEG. They are used in video-CD. Supports audio and video storage on CR-ROMS. |
| MPEG-2/H.262 | H.xxx | 1995 | Used in DVD. Supporting HDTV |
| H.263 | H.xxx | 1996 | Significant advancement in video streaming and video conferencing. Share subset with MPEG-4 |
| MPEG-4 | MPEG | 1999 | Includes DivX and Xvid. Played crucial contributions in the pre-HD era. |
| MPEG-4/H.264 | H.xxx | 2003 | It supports Blue-ray, HD DVD, and Digital video broadcasting. Co-published with H.264 |
| HEVC/H.265 | H.xxx | 2013 | Live HD streaming of the data. |
| VVC | MPEG | 2020 | Live HD streaming, OTT, etc. |

### 5.2. Traditional Approach

The process of data compression is completed by a set of encoders and decoders called codecs. Figure 20 explains the process of compression used by traditional codecs. The main objective of the codec is to identify and remove temporal and spatial redundancies from the video [64]. The transform block from the process converts a video to a series of images, and a quantizer block will encode the minimized form. Then, the entropy coding block will apply the appropriate compression algorithm and be saved to the memory. The exact process will be in reverse order to obtain the original video file after decompression. The dictionary-based learning methods [65] have the aim of minimizing the reconstruction error for images and videos. Dictionary-based coding is a successful method with many practical implementations (as mentioned in the "Introduction" section). It initially attempts to identify the feature vector in dimensional data. Then, it starts learning the dictionary by identifying and adding dimensions of the data and providing a new representation to it. These representations will be used while reconstructing the image. Dictionary learning with sparse-based representations can be used in image inpainting, classification, and denoising applications. There are various state-of-the art methods proposed for this purpose [66,67]. The types of compression and history of compression are discussed in the Introduction section. This section focuses on issues present in the current section.

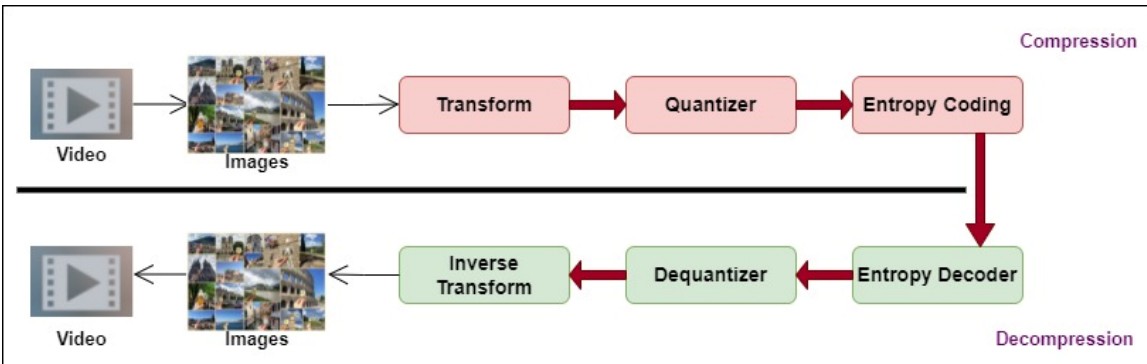

**Figure 20.** Traditional approach used by video codecs.

*5.3. Issues in the Traditional Approach*

- Traditional codecs are hardcoded approaches: All traditional codecs discussed earlier [55–61] have a static method of performing compression. Since they are specific to the input provided to the process of compression, the completed form will be disturbed when the input experiences any minor changes. Moreover, they require hand-tuning of the parameters that play a crucial role in compression.
- Traditional codecs are not adaptive: Since codecs are designed and programmed for a specific type or set of input, they cannot be used for any other kind of data. We cannot guarantee its performance to a new kind of input. This is one of the huge issues video codecs are facing, although dictionary-based learning provides adaptiveness up to its best extent.

Traditional codecs have lower compression efficiency: Most available codecs result in lower compression efficiency. Their non-adaptive nature limits their efficiency to identify redundancies in the video, resulting in a lower compression rate. The other reason that affects compression rate is their support to a lower resolution. In today's world, video data are changing its form frequently. The resolution supported by devices in the changing world is impressive. These older compression techniques cannot match the speed of changes in video formats. Moreover, they cannot be used for various new video formats such as 360 degrees AR/VR videos [68]. It has also been found that they face challenges in live video streaming [69] and coding for 3D TV [70].

- Further competition is more difficult: Because of the static and non-adaptive nature of the available video codecs, it is becoming tough to compress available data further.
- Current DNN approaches improve the rate-distortion but make the model much slower and robust. Moreover, it requires more memory which limits their practical usage.
- Today, even in the bay area, mobile network is variable. It may also cause problems in the streaming and compression of data. It is also doubtful whether the network will support high-quality data or not.

*5.4. Why Artificial Intelligence*

Artificial Intelligence (AI) is intelligence added to machines by making machines learn the set of scenarios and acquire rules or knowledge through it. It makes machines self-efficient in solving problems or in making decisions on their own. A variety of AI algorithms is currently being used in multidisciplinary fields. Machine learning (ML) and Deep Learning (DL) are particularly making advancements in several fields, and we can see a significant impact on the result of that. It shows extraordinary results in many applications by fast processing and making real-time predictions, so it is being tried for compression purposes. Video compression is challenging, and available codecs face several issues, as discussed in the last section. If we can understand and explain the main fundamentals of video compression, then machine learning may show a significant impact

on the results [71]. The following are a few reasons why Deep Neural Networks (DNN) approaches can prove themselves better in video compression:

- DL algorithms are adaptive: The beauty of DNN algorithms is their adaptiveness to the input. They learn themselves according to the input data. Even though we provide a large volume of data input, DNN algorithms can identify various trends and patterns and provide the maximum possible efficient solution to the problem. They may require extra time to learn, but they provide promising results once they understand the pattern. Moreover, humans do not need to babysit the algorithms in every execution step.
- Learn parameters to optimize compression objective: Hyperparameter tuning is crucial in generating results in DNN algorithms. Several parameters must be set at an optimum value to develop more efficient results. The adaptive nature of the DNN algorithms helps adjust those parameters as per the input given to the algorithm. Thus, programmers need not require manual calculation and manually setting those values, which reduces a significant burden on the programmer's shoulders.
- Transfer learning: Another exciting advantage DNN algorithms provide is transfer learning. Transfer learning [72] solves problems from different domains using available data and previous experiences. DNN algorithms have a comprehensive set of applications. We can try a trained model from one application to another and see whether it can provide expected results.
- Supports a variety of data: DNN algorithms support multi-dimensional and multi-variety of data. They may use ETL (Extract, transform, load) tools or tools in uncertain or dynamic environments to generate results.
- Continuous Improvement: DNN algorithms become smarter when exposed to a variety of data. They gain experiences from input data and go on improving efficiency and accuracy. Moreover, they help in increasing coding efficiency.

Although DNN algorithms show a substantial set of advantages to use, they also have challenges such as data acquisition (requires a considerable amount of data, requires data cleaning), interpretation of results, time and resources required for computing, high error susceptibility (since the system works autonomously, they are highly susceptible to errors), interpretation of errors, etc. Figure 21 have summarized the issues and advantages of DNN approaches for video compression.

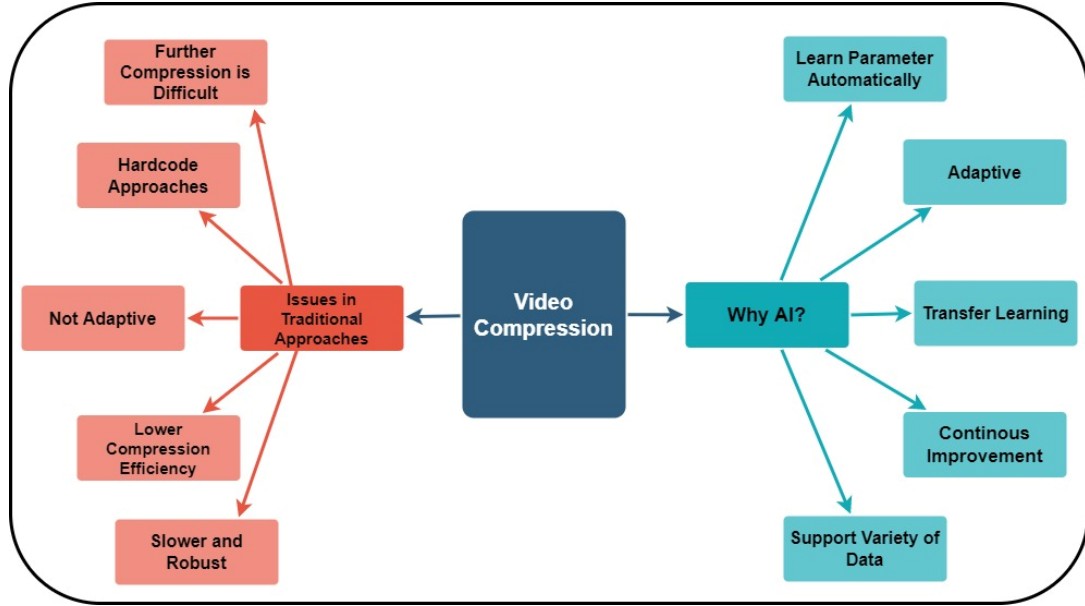

**Figure 21.** Video compression: issues and advantages of DNN approach.

*5.5. Proposed Deep Learning Approaches for Video Compression*

All video compression approaches discussed earlier in the 'History of video compression' section used a sequence of transforms and quantizers for compression. This section will briefly discuss various deep learning-based approaches used for video compression since 2020.

We have witnessed several ML-based approaches in the last decade or more. Most of them use DNN-based algorithms. DNN approaches are more powerful since they have several epochs (depending on the quantity and complexity of data) that update hyperparameter, which help train the model. It will be made ready to fetch real-world data. Evolution in techniques used in DNN approaches is represented by timeline in Figure 22. We have noticed many successful DNN based approaches for image compression [73–83]. Since they use highly nonlinear transforms and end-to-end training strategy, DNN-based approaches for image compression are proven to be very successful. The same methods are also tried for video compression [43,44,84–86], and they are successful up to some extent [50,87–91]. The following are DNN-based video compression approaches, and the latest is from 2018. Also, Table 19 have distinguished approaches according to type of compression they are doing. Guo lu et al. [48] proposed the first end-to-end video compression system. All other traditional methods modified either one or two modules from the compression process. The best part of this system is that it takes advantage of classical compression architecture and a neural network of non-linear representation. This algorithm outperforms the performance of both H.264 and H.265.

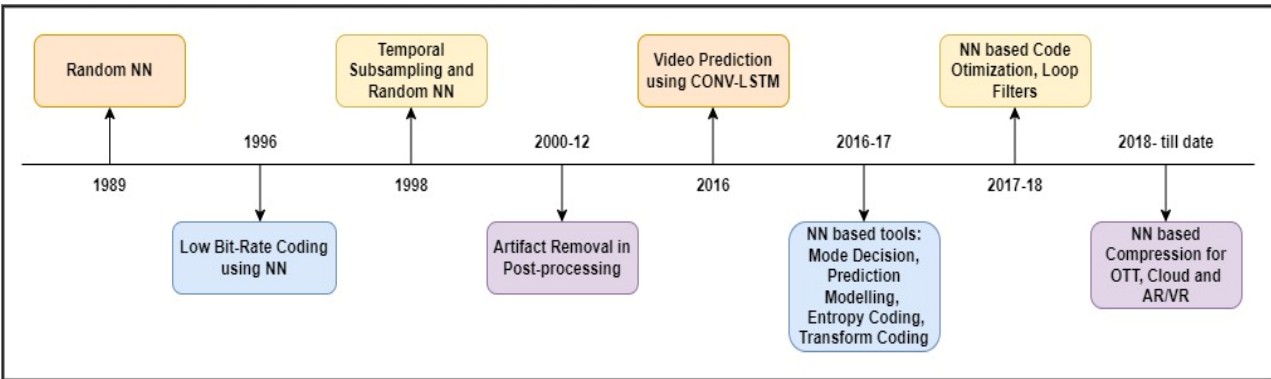

**Figure 22.** Timeline for DNN based video compression.

DNNs are very effective in computer vision tasks, but they are vulnerable to adversarial attacks. Therefore, the study of defense against adversarial attacks is critical. Adversarial attacks are of two categories: white-box and black-box [92]. In white-box attacks, the adversary has direct access to the model. In black-box attacks, adversaries have limited access to models. These attacks are possible in videos also. Wei et al. [93] explained that attacking a few frames from the video will confuse the model and produces the wrong results. These attacks can also identify action recognition from videos [94]. Yupeng Chen et al. [95] proposed a two-stage framework to prevent the model from such attacks. Self-adaptive JPEG compression provides an efficient compression, and Optical Texture-based Defense (OTD) controls the optical flow of frames to suppress chances of adversarial attacks on it.

**Table 19.** A proposed approach for types of compression.

| Types of Compression and Proposed Approaches | |
| --- | --- |
| **Lossy** | **Lossless/Near Lossless** |
| Guo lo et al., [48] | Darwish et al., [96] |
| Yupeng Chen et al., [95] | Wei Jia et al., [97,98] |
| Sangeeta et al., [99] | Ghamsarian, N. et al., [37] |
| Woongsung Park et al., [100] | Sinha, A.K. et al., [101] |
| Dhungel P et al., [102] | Santamaria M et al., [103] |
| Zhu S et al., [104] | Ma D et al., [105] |
| Poyser M et al., [106] | He G et al., [107] |
| Mameli F et al., [108] | Feng R et al., [109] |
| Chen W et al., [110] | Liu D et al., [111] |
| Pham T et al., [112] | Chen Z et al., [113] |
| Jadhav A et al., [114] | Wu Y et al., [115] |
| Lu G et al., [116] | Ma D et al., [117] |

Darwish et al. [96] proposed an optimized video codec that adapts the genetic algorithm to build an optimal codebook for adaptive vector quantization. This will be used as an activation function in neural networks. A background subtraction algorithm is used to extract motion objects from the frame. It helps in generating a context-based initial codebook. Differential Pulse Code Modulation (DPCM) is applied for the lossless compression of significant wavelet coefficients. Learning Vector Quantization (LVQ) neural networks are used for the lossy compression of low energy coefficients. In the final step, Run Length Encoding (RLE) is employed to achieve a higher compression ratio. Experimental results prove the efficiency of the system. PSNR is a metric used for performance analysis.

Augmented reality and virtual reality are evolving applications [118]. The point cloud is a format used in 3D object modeling and interactions in those applications. Wei Jia et al. [119] proposed a self-learning-based system to remove geometry artifacts to improve compression efficiency in Video-Based Point Cloud Compression (V-PCC). This is the first approach perform this process. It shows promising results to remove geometric artifacts and reconstruct 3D videos. Another method is proposed by Wei Jia et al. [98], who is offering to improve the accuracy of occupancy map video using CNN.

Sangeeta et al. [99] proposed a video compression technique based on ConvGRU, a convolutional recurrent neural network that combines the advantages of both CNN and RNN. The randomized emission step ConvGRU-based architecture used in the system results in better performance and can be helpful in further optimization enhancements.

Woongsung Park et al. [100] proposed DeepPVC, an end-to-end deep predictive video compression network. The CNN-based approach outperforms AVC and HEVC as it performs parallelly decoding video data.

Moreover, in 2020, several approaches were proposed using DNN techniques. Table 19 provides detailed information about it. Table 20 is a brief summary of method used for compression, datasets, and their real time applications. The study shows that CNN is a widely used image or video compression technique. Few researchers have tried using Generative Adversarial Networks (GAN), and a few have used Recurrent Neural Networks (RNN) for this purpose. Autoencoder (AE) is also being preferred for compression purposes. Figure 23 summarizes the various technologies used for video compression.

**Table 20.** Video compression approaches using DNN.

| Document | Method of Compression | Dataset Used | Application |
|---|---|---|---|
| **Guo lo et al., [48] 2021** | CNN | UVG, HEVC | OTT, Video Steaming |
| **Yupeng Chen et al., [95] 2021** | Long-term recurrent convolutional networks (LRCN) | UCF101 | Optical texture preservation in compression |
| **Darwish et al., [96] 2021** | Differential Pulse Code Modulation (DPCM), Learning Vector Quantization (LVQ) | xiph.org | Video Streaming and transmission |
| **Wei Jia et al., [98,119] 2021** | Video-Based point cloud compression (V-PCC), CNN | CTC | Point cloud for 3-D object modeling, AR and VR |
| **Sangeeta et al., [99] 2021** | RNN, CNN | | OTT, social media, Storage for online video content |
| **Woongsung Park et al., [100] 2021** | CNN | UVG, HEVC-B, HEVC-E | Storage for online video content |
| **Dhungel P et al., [102] 2020** | DNN | UVG, HEVC | Storage for online video content |
| **Ghamsarian, N. et al., [37] 2020** | CNN | Medical Dataset- Cataract-101 | Medicine Videos-Cataract Surgery |
| **Sinha, A.K. et al., [101] 2020** | CNN | UVG, Kinetic 5K | Live streaming. broadcasting |
| **Santamaria M et al., [103] 2020** | DNN | DIVerse 2K (DIV2K) | Videos with High Resolution |
| **Ma D et al., [105] 2020** | GAN | HEVC | Spatiotemporal data |
| **Zhu S et al., [104] 2020** | CNN | HEVC | Spatiotemporal data |
| **He G et al., [107] 2020** | ORNN | CLIC | CVF Competition |
| **Feng R et al., [109] 2020** | DNN | Vimeo-90K, CLIC | CVF Competition |
| **Liu D et al., [111] 2020** | HEVC, VVC | CNN | Real-time videos |
| **Chen Z et al., [113] 2020** | Flicker | PMCNN | Social Media |
| **Poyser M et al., [106] 2020** | R-CNN, GAN, encoder | Cityscapes | Real-time videos |
| **Mameli F et al., [108] 2020** | No-GAN | | Real-time videos |
| **Wu Y et al., [115] 2020** | RNN, GAN | Surveillance data | Surveillance video applications |
| **Chen W et al., [110] 2020** | CNN | JCT-VC | HD Videos |
| **Pham T et al., [112] 2020** | CNN | HMII | Video Streaming and conferencing |
| **Ma D et al., [117] 2020** | CNN | BVI-DVC | Video Streaming and conferencing |
| **Jadhav A et al., [114] 2020** | PredEncoder | Youtube Videos | Video Streaming and conferencing |
| **Lu G et al., [116] 2020** | DNN | Vimeo-90K, HEVC | Video Streaming and conferencing |

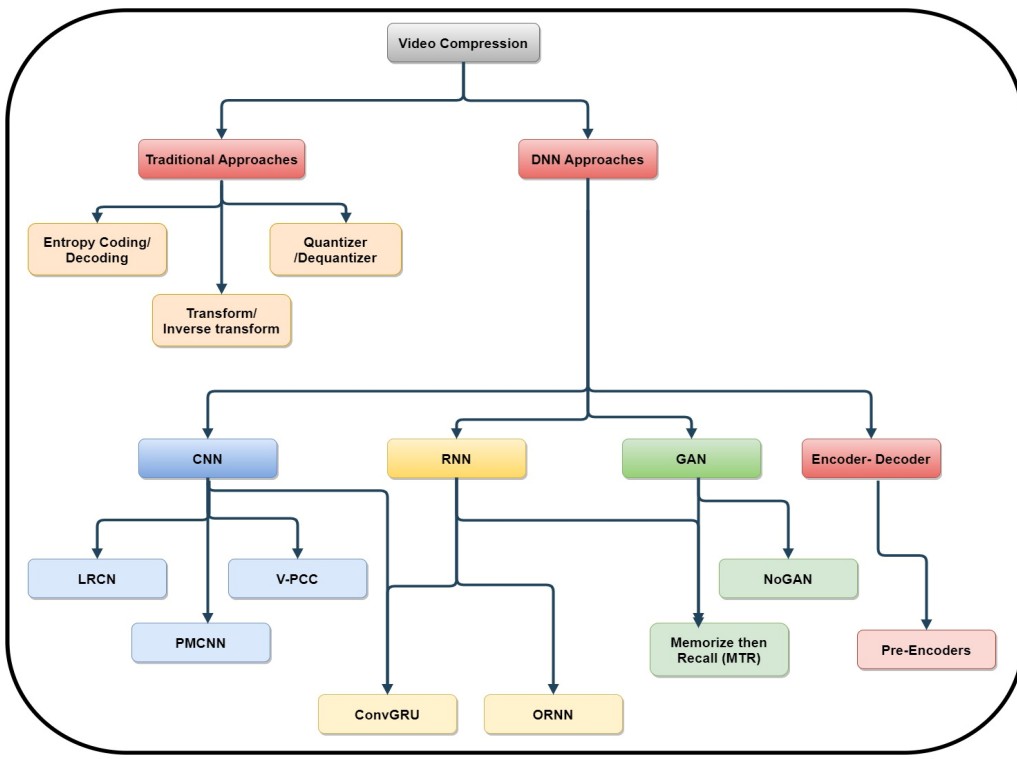

**Figure 23.** Video compression technologies.

*5.6. Metrics for Performance Measurements*

Videos undergo a series of processes before being displayed to the world. As discussed in an earlier section, initially, the video is converted to a series of images. Then, the images undergo processing. This processing will work on images, which may affect images in one way or another. These changes introduce new artificial artifacts in the picture and degrade its quality. The image degradation may include blurriness, geometric distortion, and blockiness artifacts because of compression standards. Therefore, measuring the quality of images/videos is an essential aspect of data reduction or data compression. Figure 24 introduce us to the various metrics who are frequently for images and videos. Table 21 explains performance metrics used by the famous video compression approaches.

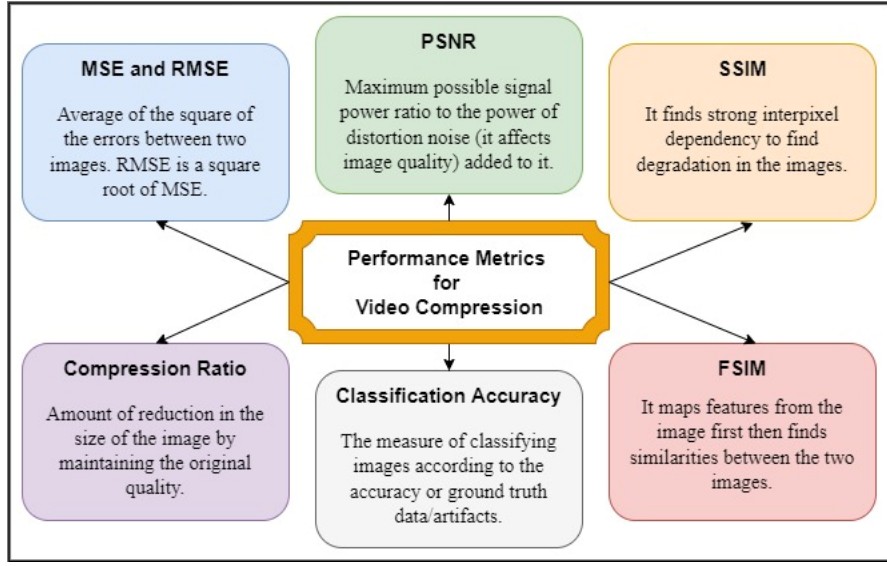

**Figure 24.** Performance metrics for video compression.

**Table 21.** Approaches and performance metrics used.

| Document | RSME | PSNR | MS-SSIM | BD-Rate | CA | CR | Performance |
|---|---|---|---|---|---|---|---|
| **Guo lo et al., [48] 2021** | | √ | √ | | | | PSNR gain= 0.61 dB |
| **Yupeng Chen et al., [95] 2021** | | | | | √ | | CA = 0.9311 |
| **Darwish et al., [96] 2021** | | √ | | | | √ | CR = 5.94% improvement |
| **Wei Jia et al., [98,119] 2021** | | | | √ | | | Significant gain in 3-D artifact removal and time complexity. |
| **Woongsung Park et al., [100] 2021** | | √ | √ | | | | MS-SSIM for HEVC-E class = 0.9958 |
| **Dhungel P et al., [102] 2020** | | √ | √ | | | | for UVG dataset MS-SSIM = 0.980 PSNR = 38 DB |
| **Ghamsarian, N. et al., [37] 2020** | | √ | | | | | Up to 68% storage gain |
| **Sinha, A.K. et al., [101] 2020** | | √ | √ | | | | Up to 50% improvement in encoding time |
| **Santamaria M et al., [103] 2020** | | | | √ | | | Improvement in BD Rate |
| **Ma D et al., [105] 2020** | | √ | √ | | | | Bit rate saving up to 24.8% |
| **Zhu S et al., [104] 2020** | | √ | √ | √ | | | 2.59 times higher efficiency than MQP |
| **Feng R et al., [109] 2020** | | | √ | | | | MS-SSIM = 0.9968 |
| **Mameli F et al., [108] 2020** | | | √ | | | | SSIM = 0.5877 |
| **Wu Y et al., [115] 2020** | | √ | √ | | | | MS-SSIM = 0.82, PSNR = 25.69 db |
| **Chen W et al., [110] 2020** | √ | √ | √ | | | | PSNR = 43 dB MS-SSIM = 0.99 |
| **Pham T et al., [112] 2020** | | √ | √ | | | | PSNR Gain = 0.58 dB |

1. MSE (Mean Square Error): It is the most common, simplest, and most widely used method of assessment of the quality of the image. This method is also called Mean Squared Deviation (MSD). This method calculates the average of the square of the errors between two images. The following is the detailed formula [120] for MSE or MSD. A value closer to zero is a measure of the excellent quality of the image. MSE between two images such as $\vec{a}(x, y)$ and $\vec{b}$.

$$MSE = \frac{1}{pq} \sum_{r=0}^{p} \sum_{s=1}^{q} \left[ \vec{b}(r, s) - \vec{a}(r, s) \right]^2$$

2. RMSE (Root Mean Square Error): This is another method to assess the quality of the image. RMSE can be calculated by taking the square root of MSE. It is an accurate estimator of errors between images. The following is the formula [120].

$$RMSE\left(\vec{\beta}\right) = \sqrt{MSE\left(\vec{\beta}\right)}$$

3. PSNR (Peak Signal to Noise Ratio): Various processes add noise distortion to the video/image. PSNR [121] measures the maximum possible signal power ratio to the power of distortion noise added to it. It is the most widely used method for assessing the quality of images after lossy compression by the codec. The following is the formula [120]. Here, peakval (Peak Value) is the maximal in the image data. If an 8-bit unsigned integer data type occurs, the peakval is 255.

$$PSNR = 10 \log_{10} \left[ \frac{peakval^2}{MSE} \right]$$

4.  SSIM (Structure Similarity Index Method): It is one of the very well-known methods of calculating image degradation [121]. This method finds strong interpixel dependency to find degradation in images. Luminance, contrast, and structure are the factors considered in finding structural similarities between images. Multi-Scale Structural Similarity Index Method (MS-SSIM) is the advanced version of SSIM. It is used to evaluate the various structural similarity of the images of different scales. The size and resolution of images are extra factors considered compared to SSIM. A three-component SSIM (3-SSIM) [122] is a newly proposed method based on Human Visual systems (HVS). A human eye can observe the difference between various textures more efficiently than any system; this advantage is used in this method. We can also calculate dissimilarity between two images; we call it a DSSIM (Structural Dissimilarity). The following is an equation outlining the calculation of SSIM [123] and DSSIM [120].

$$\text{SSIM } (\vec{A}, \vec{B}) = \left[ x\,(\vec{A}, \vec{B}) \right]^{\alpha} \cdot \left[ y\,(\vec{A}, \vec{B}) \right]^{\beta} \cdot \left[ z\,(\vec{A}, \vec{B}) \right]^{\gamma}$$

The luminance of the image can be calculated by the following.

$$x\,(\vec{A}, \vec{B}) = \frac{\left[ (2\,p_{\vec{A}}\,p_{\vec{B}}) + C_1 \right]}{(p_{\vec{A}}^2 + p_{\vec{B}}^2 + C_1)}$$

The contrast of the image can be calculated by the following.

$$y\,(\vec{A}, \vec{B}) = \frac{\left[ (2\,q_{\vec{A}}\,q_{\vec{B}}) + C_2 \right]}{(q_{\vec{A}}^2 + q_{\vec{B}}^2 + C_2)}$$

The structure of the image can be calculated as follows:

$$z\,(\vec{A}, \vec{B}) = \frac{\left[ \sigma(\vec{A}, \vec{B}) + C_3 \right]}{(\sigma(\vec{A})\sigma(\vec{B}) + C_3)}$$

where $p_{\vec{A}}$ and $p_{\vec{B}}$ are local means, $q_{\vec{A}}$ and $q_{\vec{B}}$ are standard deviations, and $\sigma(\vec{A}, \vec{B})$ is the cross-covariance for images $\vec{A}$ and $\vec{B}$, respectively. If $\alpha$, $\beta$, and $\gamma$ are equal to 1, then the index is simplified as follows.

$$\text{DSSIM } (\vec{A}, \vec{B}) = \frac{1 - \text{SSIM } (\vec{A}, \vec{B})}{2}$$

5.  Features Similarity Index Matrix (FSIM): FSIM [124] is an advanced method that maps features from the image first and then finds similarities between two images. The method of mapping features in the picture is called Phase Congruency (PC), and the method of calculating the similarity between two images is called Gradient magnitude (GM).
6.  Classification Accuracy (CA): It is again one of the measures for the classification of images. This method compares the generated image with the original image and declares how accurate it is. It uses a few sampling methods to do so. Accuracy can be based on data available of the original image, which sometimes may be collected manually, so it may be a time-consuming process.
7.  Compression Rate (CR): It is a measure that explains what percentage of the original image is compressed without losing essential or important contents/artifacts of the image. It is widely used in applications such as photography, spatiotemporal data, etc.

### 5.7. Study of Datasets

Datasets are distributed according to the color space they possess. The color space defines the organization of color in the image. Various hardware devices can support the different representations of the color; maybe it differs because of analog or digital replicas of the data on the screen. Color space supports various color models. A color model is a mathematical model that explains how colors can be represented on the screen. The color models used for the distribution of the dataset are RGB (color space of values representing red, green, and blue color) and YUV (color space with luma component (Y'), blue projection (U), and red projection (V)). Figure 25 is a consolidated summary of available datasets according to their color space.

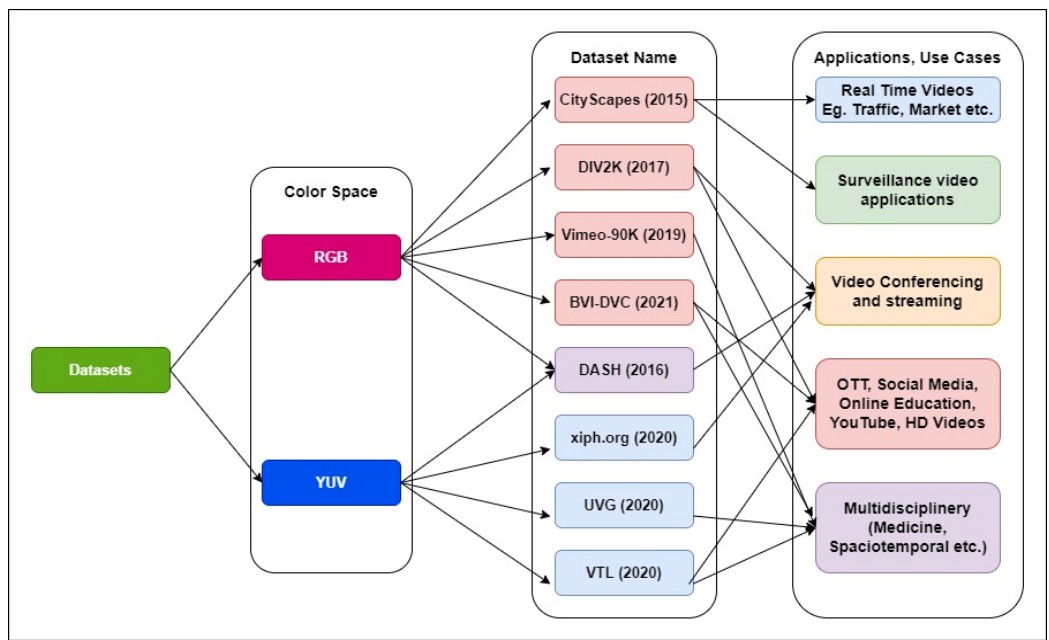

**Figure 25.** Datasets used in video compression with a year of introduction.

Dynamic Adaptive Streaming over HTTP (DASH) [125–127] is a very famous dataset specially designed for AVC (H.264) and HEVC (H.265) video codecs. It supports YUV color space. This dataset contains trace-based simulation videos of NS-2 and NS-3, and the testbed simulation dataset available is used for analyzing the delivery of data contents over a physical network. This dataset provides the liberty of adapting according to client mechanisms.

Another very famous dataset widely used for video compression is CityScapes [128,129]. It supports RGB color spaces. This dataset has recorded videos of city scenes from different 50 cities. It provides diversified data from various seasons (spring, summer, and fall) and data with 30 other classes and a variety of annotations.

Video Trace Library (VTL) [130–132] is another dataset with support of YUV color space. This dataset provides traces of H.263, H.264, MPEG-4, wavelets, and pre-encoded internet content. The focus of this dataset is for traffic monitoring and the prevention methods. Thus, it provides recorded videos of traffic with vehicles of a variety of classes.

Ultra Video Group (UVG) [133] contains 16 versatile 4K (3840 × 2160) test video sequences. They provide natural sequences recorded at 50 to 120 FPS (frames per second) and are stored in raw 8-bit and 10-bit YUV format. It is specifically designed for the subjective and objective assessment of our new generation codecs.

Xiph.org [132] is another famous source to obtain a video dataset. It provides a dataset supporting YUV 4:4:4 and YUV 4:2:0 format. A variety of videos with different bit rates, classes, and duration is available on the referred site.

Diverese2K (DIV2K) [133–135] is a new dataset supporting RGB color space. CVPR (Computer Vision Premium Event) [134] is an annual event with the central theme of

computer vision. NTIRE (New Trends in Image Restoration and Enhancement workshop) is a conference cum workshop organized in the theme of image/video restoration and image/video enhancement. For papers related to mentioned topics, organizations have released the DIV2K dataset as a sample dataset to simulate the results. NTIRE 2017, NTIRE 2018, and PRIM 2019 are organized under this initiative.

BVI-DVC [135] is one of the latest datasets released with the color space of RGB. This dataset is made available primarily for deep video compression. It provides 800 video sequences from 270p to 2160p. It is made available primarily for CNN-based video compression tools, aiming to enhance conventional coding architectures.

Vimeo-90K [136] is another famous video dataset. It supports RGB color space. Around 90,000 videos are made available on vimeo.com (4 January 2022). These videos are of a variety of scenes of activities. It is designed mainly for four video processing tasks: temporal frame interpolation, video denoising, video deblocking, and video super-resolution.

## 6. Discussion

In qualitative analysis, we discussed issues, advantages, and disadvantages of ML algorithms, performance metrics, available datasets, and proposed approaches for video compression. The conclusion we can make is that there are many expectations from data compression algorithms in the case of images and videos. The user wants to save its physical as well as virtual space. Moreover, a user is expecting high-quality data back after decompression. A few approaches satisfy the need for some applications but are challenging for other applications. Thus, as mentioned above, we need to provide a very efficient data compression system that will guarantee optimum performance in all different kinds of applications. The high-performing codec can be helpful in applications such as OTT, video streaming, video conferencing, digital television broadcasting [137–139], social networks, the field of medicine, field of agriculture, wireless sensor networks, etc. Moreover, this section will discuss the top applications of video compression; challenges we are currently facing in video compression; competitions; and information of events supporting research in video compression.

### 6.1. Challenges in Video Compression

We are currently experiencing innovations in video capturing, storing, and display technologies every year. Thus, matching the speed of innovations is also a considerable challenge for video codecs. Every user wants to have a great experience at their end. Still, they do not know the efforts video codecs have to put in behind delivering it, as well the complexity of the operation of producing, storing, and providing the data to the user. The following are the challenges faced by today's video codecs, also explained by Figure 26.

- Faster encoders do not guarantee potential compression efficiency: Most codecs try their best to compress, but they do not promise us a potential compression efficiency. Although few can perform good compression levels, they are slower than older codecs. It may be because of the variety and complexity of data being generated by devices today. The other reason may be the data formats of the data; they are changing very quickly. HFR, HDR, 4K, 6K, 8K, 3D, and 360-degree videos are newly evolved challenging formats.
- Encoder search problem: Finding an efficient encoder for data compression is challenging. There are several hurdles in the path that must cross by the encoder. Currently, ML algorithms are extensively being used to reduce the complexity of the encoder. However, we must admit that ML has its advantages and disadvantages.
- Many software encoders can support lower resolutions: Compression is becoming more difficult because of changes in the resolution of the data. It is becoming tough to find redundancy between data and further compression.
- Further compression is more complex: Obtaining an efficient output depends on the changes made in the ML model and the hardware required to run that model. They are time consuming and costly. Input given to the data is not really in the programmer's

hands; it will change in the future, so we need to develop a system that will adapt to the changes and help us do different level compression.

- Deep learning methods are very successful for applications such as image classification. However, the system was found to be very instable when it comes to image restoration. A tiny change occurring on the image results in losing artifacts or important features from the image. It also led to a degradation of the quality of the image. This may occur because of changes in resolution, faulty source equipment, use of inappropriate method for processing the image, etc. This instability makes us think about how much we should rely of these deep learning-based methods.

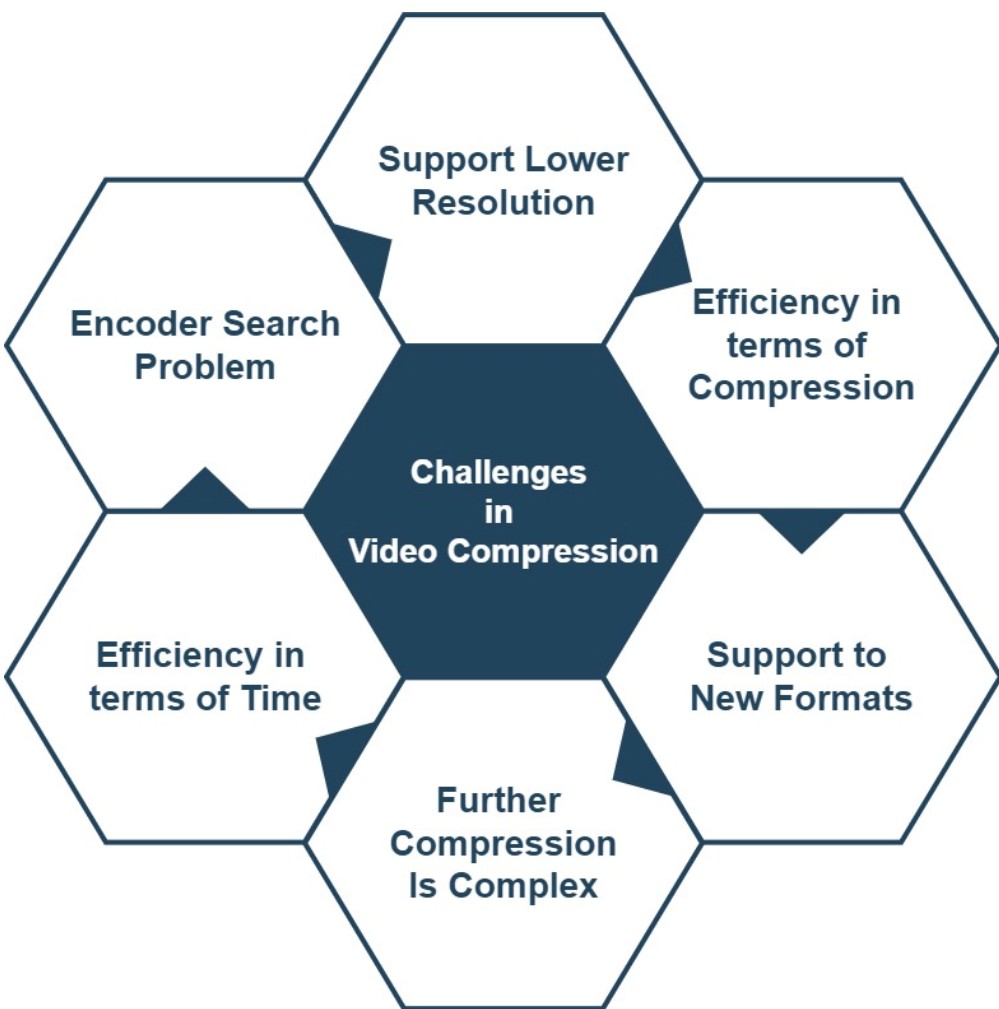

**Figure 26.** Challenges in video compression.

*6.2. Important Findings from the Analysis*

After performing quantitative and qualitative analyses, the following important findings are listed:

- It has been observed that a considerable amount of compression work has been performed for textual data. Since there are a limited number of characters available in every language and no new formats are expected in the future, we can conclude that data compression for textual data has almost reached an end. However, most of the work can be performed on the encryption and decryption parts based on the requirement of an application. However, this is not the case for multimedia data, especially images and videos. After performing a bibliometric quantitative and qualitative studies on images, it has been found that a lot of work is completed

or ongoing to achieve efficient image compression. The latest compression work is adapting to new formats evolving for the images. However, when a bibliometric study has been performed for video data, it has been found that a tremendous amount of work is ongoing for video data, and it is trying to match the growth of internet-scaled video traffic.

- International Telecommunication Union (ITU), International Standards Organization (ISO), and International Electrotechnical Commission (IEC) are major organizations that are working in the domain of video compression. MPEG and H.xxx are two families proposed by them. Versatile Video Coding (VVC) is the latest approach proposed by them in 2020. It has good results in live HD streaming and other online platforms.
- Traditional codecs were using a set of transformers (and inverse transform) and quantizers (and dequantizer) for video compression. The main issue with them is the hardcoded approach. It requires hand-tuning the parameters. Moreover, these approaches were static, so they are not adaptive and provide a lower compression rate.
- Using the DNN-based approach is a solution to issues with the traditional approach. They are adaptive, support a variety of data, and provide a promising compression rate. They support transfer learning and show continuous improvement in learning the data and providing results.
- A variety of DNN approaches was used for the image as well as video compression. CNN is a widely used approach. RNN, GAN, encoders, and ensembled methods are the current approaches favored by researchers. They are widely being used in a variety of applications such as OTT, social media, online education, surveillance systems, live streaming of HD data, video conferencing, and various multidisciplinary fields.
- PSNR (Peak Signal to Noise Ratio), SSIM (Structure Similarity Index Method), classification accuracy, and compression rate are metrics used for the performance analysis.
- Many video datasets are freely available to access. CityScapes, DIV2K, UVG, and xiph.org are a few famous datasets that are used by researchers. For applications in healthcare or space or surveillance systems, datasets need to be generated or should be made available by government institutes/organizations for testing purposes.
- The computer vision foundation (CVF) [133] is a nonprofit organization that promotes and supports research in the field of computer vision. It organized three kinds of events named CVPR (Computer Vision and Pattern Recognition), ICCV (International Conference on Computer Vision), and ECCV (European Conference on Computer Vision). Through these events, various workshops, courses, and competitions are organized. It also publishes research in the domain of computer vision. New Trends in Image Restoration and Enhancement (NTIRE) include the workshop and challenges on image and video restoration and enhancement organized by CVPR conferences. Advances in Image Manipulation (AIM) are workshops challenges on the photo and video manipulation organized by ECCV conferences.
- Alliance of Open Media (AOM) [134] is a famous organization that developed AV1 and AVIF. It has started investigating next-generation compression technology. It has challenged the world to design codecs beyond AV1.
- Stanford Compression Forum (SCF) [33] is a research group that extensively supports and promotes research in data compression. A group of researchers from Stanford University started this forum. This forum aims to transform academic research into technology or timely research problems or provide training in the field of data compression. "Stanford Compression Workshop 2021" is the latest event organized by this forum in February 2021.

### 6.3. Future Directions

It is essential to identify an intelligent model governing the data for many real-time applications, such as making predictions based on data or understanding its causal processes. For example, in the case of video calling, it may require seeing the person or people then other objects in the frame. Another example involves a tennis match. In a tennis match,

it is more important to preserve the quality of players and the court than the distinguishing feature of the crowd. Information theory tells us that a good predictor forms suitable compressors. In such cases, the use of machine learning approaches meets our expectations. Countless machine learning algorithms perform functions such as regression, classification, clustering, decision trees, extrapolation, and more. Machine learning trains algorithms to extract the data's information to perform a data-dependent task. While designing such algorithms, various machine learning approaches such as Supervised Learning, Unsupervised Learning, Reinforcement Learning, etc., can be used [44]. Available DNN approaches improve the rate-distortion, making the model much slower and more robust. Moreover, it requires more memory, which limits their practical usage. Researchers may focus on this issue while proposing new approaches.

**Author Contributions:** Conceptualization, R.V.B., S.M. and S.P.; methodology, R.V.B., S.M. and S.P.; software, R.V.B., S.M., S.P. and B.Z.; validation, R.V.B., S.M., S.P., K.S., D.R.V., K.K. and B.Z.; formal analysis, R.V.B. and B.Z.; investigation, R.V.B. and B.Z.; resources, R.V.B. and B.Z.; data curation, R.V.B., S.M., S.P., K.S., D.R.V., K.K. and B.Z.; writing—original draft preparation, R.V.B.; writing—review and editing, R.V.B., S.M., S.P., K.S., D.R.V. and K.K.; visualization, R.V.B. and B.Z.; supervision, S.M., S.P., K.S., D.R.V. and K.K.; project administration, S.M., S.P., K.S., D.R.V. and K.K. All authors have read and agreed to the published version of the manuscript.

**Funding:** This research received no external funding.

**Institutional Review Board Statement:** Not applicable.

**Informed Consent Statement:** Not applicable.

**Data Availability Statement:** Not applicable.

**Conflicts of Interest:** The authors declare no conflict of interest.

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
