# Peer review of "Deep Learning Approaches for Video Compression: A Bibliometric Analysis"

_2504-2289, doi:10.3390/bdcc6020044_

Round 1
Reviewer 1 Report
Thank you for submitting the paper. The paper performed a thorough overview of conventional methods and deep learning-based video compression techniques, which can be the basis for future studies. The topic is important and needed with the growing video conferences. However, there are many language issues and I cannot fully understand the paper. In addition, the main argument of this paper must be reflected. Please reorganize it and revise the paper carefully before resubmission.
General comments:
1. The definition of the conventional method is unclear. It seems that the author assumes that "conventional methods" are always non-adaptive. A counterexample to their argument is "dictionary learning", which is a fast adaptive conventional method. Please discuss the role of fast adaptive dictionary learning in video compression.
2. The main concern about the DL method in image recognition is instability. These may degrade the image quality in some cases.
Major issues:
Generally there exist many language issues. The manuscript must be edited and proofread by an experienced English writer.
Otherwise, readers cannot understand the paper. Below is a non-exhausted list of errors.
1. Incorrect upper-case letters. For example: in Line 30 "the Advantages" should be "advantages"; in Line 39 Adaptability; in Line 853, etc.
2. Line 864: Or, in a tennis match, it is more important to preserve the quality of players and the court than the distinguishing feature of the crowd. The sentence is not suitable for Future directions.
3. There exist too many grammar errors. e.g. in Line 26: Scopus and Web of Science are well-known research databases are used for this analytical study.
4. Fonts in figures are too small.
5. Repeated references 142, 143.
6. Ref 2: ben should be Ben
7. Other issues.
Author Response
Dear Reviewer, Thank you for the detailed review. I have tried making all possible changes as per the comments.

Reviewer 2 Report
- The number of pages in the manuscript that are too long needs to be shortened, but it still ensures to keep the vital information and logic of the article.
- Some figures are blurred, and the author needs to correct and provide a higher resolution, for example, Figures 1, 3, 4, 21, 22, etc.
- Some tables are too long, need to shorten or remove the title of the article in the table presentation
Author Response

(The authors gave the same response as above.)

Round 2
Reviewer 2 Report
Accept for publication